EMBO
reports

# mRNA nuclear retention reduces AMPAR expression and promotes autistic behavior in UBE3A-overexpressing mice

Yuan Tian [ID] [1], Feiyuan Yu[1], Eunice Yun [ID] [1], Jen-Wei Lin[1] & Heng-Ye Man [ID] [1,2,3 ✉]

## Abstract

*UBE3A* is a common genetic factor in ASD etiology, and transgenic mice overexpressing UBE3A exhibit typical autistic-like behaviors. Because AMPA receptors (AMPARs) mediate most of the excitatory synaptic transmission in the brain, and synaptic dysregulation is considered one of the primary cellular mechanisms in ASD pathology, we investigate here the involvement of AMPARs in UBE3A-dependent ASD. We show that expression of the AMPAR GluA1 subunit is decreased in UBE3A-overexpressing mice, and that AMPAR-mediated neuronal activity is reduced. GluA1 mRNA is trapped in the nucleus of UBE3A-overexpressing neurons, suppressing GluA1 protein synthesis. Also, SARNP, an mRNA nuclear export protein, is downregulated in UBE3A-overexpressing neurons, causing GluA1 mRNA nuclear retention. Restoring SARNP levels not only rescues GluA1 mRNA localization and protein expression, but also normalizes neuronal activity and autistic behaviors in mice overexpressing UBE3A. These findings indicate that SARNP plays a crucial role in the cellular and behavioral phenotypes of UBE3A-induced ASD by regulating nuclear mRNA trafficking and protein translation of a key AMPAR subunit.

**Keywords** ASD; AMPAR; mRNA Nuclear Export; SARNP; Synaptic Transmission
**Subject Categories** Molecular Biology of Disease; Neuroscience

## Introduction

Autism spectrum disorder (ASD) is a complex neurodevelopmental disorder characterized by challenges with social interaction and communication, along with restricted and repetitive behaviors (Lord et al, 2020). The presentation of these impairments varies in range and severity, and they often co-exist with other disorders, such as attention-deficit/hyperactivity disorder (ADHD), anxiety, depression, and epilepsy (Lord et al, 2020). ASD is prevalent in about 1% of the global population, and this number continues to increase (Maenner et al, 2021). Both genetic variations and environmental factors in early development have been shown to contribute to the pathogenesis of ASD (Abrahams et al, 2013; Sandin et al, 2012; Estes and McAllister, 2016; Boukhris et al, 2016). Although the etiological causes of ASD are heterogeneous, ASD patients show common features such as abnormalities in brain growth during early childhood (Wegiel et al, 2010), often accompanied by abnormal neuronal activity and connectivity (Van Kooten et al, 2008; Bailey et al, 1998; Khatri et al, 2018; Yin and Schaaf, 2017). Co-expression network analyses of ASD-linked genes suggest that early postnatal neurodevelopmental processes in neurogenesis, neuron migration and maturation, and synaptic formation are key vulnerability points in ASD (Parikshak et al, 2013).

The synapse is considered a primary cellular substrate in ASD pathology, demonstrating aberrance in synaptic transmission, excitatory-inhibitory (E/I) ratio, and overall neuronal network connectivity (Gilbert and Man, 2017; Bourgeron, 2009; Zoghbi and Bear, 2012; Phillips and Pozzo-Miller, 2015; Bagni and Zukin, 2019; Redies et al, 2012; Lee et al, 2017). The understanding that functional abnormalities at the synapse are a common pathology in ASD was first gained from the phenotypic overlap of ASD with similar monogenic diseases involving synaptopathy, such as fragile X syndrome and tuberous sclerosis complex (Bourgeron, 2009; Guang et al, 2018). Studies have demonstrated that mutations in numerous ASD genes converge on cellular pathways that intersect at synapses. These genes encode cell adhesion molecules, synaptic receptors, scaffolding proteins, and proteins involved in synaptic transcription, local protein synthesis and degradation, affecting various aspects of synapses including synapse formation and elimination, synaptic transmission, and plasticity. Alterations in dendritic spine density are a commonly observed phenotype in monogenetic ASDs. ASDs of *PTEN*, *TSC1*, and *SYNGAP1* have been shown to lead to an increase in spine density, whereas ASDs resulting from a loss of *NEXMIF*, *SHANK3*, *NCKAP1* and *MeCP2* often show decreased spine density (Kwon et al, 2006; Peça et al, 2011; Carlisle et al, 2008; Gilbert et al, 2020). Additionally, many ASD susceptibility genes, including *NEXMIF*, *SHANK3*, *SYNGAP1*, *NRXN1*, *NLGN3*, *CYFIP1*, and *CNTNAP2*, demonstrate

[1]Department of Biology, Boston University, 5 Cummington Mall, Boston, MA 02215, USA. [2]Department of Pharmacology, Physiology & Biophysics, Boston University School of Medicine, 72 East Concord Street, Boston, MA 02118, USA. [3]Center for Systems Neuroscience, Boston University, 610 Commonwealth Avenue, Boston, MA 02215, USA.
✉E-mail: hman@bu.edu

abnormalities in synaptic function such as alterations in excitatory and inhibitory synaptic currents (Peça et al, 2011; Bozdagi et al, 2010; Sacai et al, 2020; Gilbert et al, 2020). These findings underscore a key role for synaptic dysregulation in ASD.

*UBE3A* is a significant ASD gene located on chromosome 15q11.2. ASD diagnosis due to genetic alterations in *UBE3A* accounts for 1-2% of all autism cases, making it one of the strongest genetic risk factors for autism (Abrahams and Geschwind, 2008; Hogart et al, 2010). The protein product of *UBE3A* gene functions as an E3 ubiquitin ligase regulating protein ubiquitination and degradation, and as a transcription coactivator in the nucleus mediating gene expression (Khatri and Man, 2019). It is well established that deletion of the *UBE3A* gene results in Angelman syndrome (Greer et al, 2010). On the contrary, excessive UBE3A expression and/or higher UBE3A activity are implicated in human ASD, often resulting from duplication or triplication of maternal chromosome 15q11-13 (Finucane et al, 2021; Khatri and Man, 2019). *UBE3A* is genetically imprinted in neurons, meaning that only the maternal copy of the gene is expressed, and the paternal copy is suppressed (Khatri and Man, 2019). Consequently, a 15q duplication at the maternal chromosome would increase the gene dosage of *UBE3A*, whereas a duplication at the paternal chromosome would have no effect (Chamberlain and Lalande, 2010). Consistently, individuals with maternal duplication of 15q11-13 experience complete penetrance of the associated syndrome, while those with paternal duplications typically show little to no effect (Cook et al, 1997; Urraca et al, 2013). Because *UBE3A* is the only gene within the 15q11-13 region consistently expressed solely from the maternal allele in mature neurons (Albrecht et al, 1997), it strongly suggests an implication of *UBE3A* in ASD. Additionally, microduplication of chromosome 15q11.2 that encompasses only the *UBE3A* gene (Noor et al, 2015), or de novo hyperactivating missense mutations of *UBE3A* (Yi et al, 2015; Iossifov et al, 2014; Weston et al, 2021) are also implicated in human ASD patients. To recapitulate the human conditions, transgenic mice with two extra copies of genomic *Ube3a* (*Ube3a* 2xTg) display typical autistic features including defective sociability, impaired communication, and increased repetitive self-grooming behavior (Smith et al, 2011; Krishnan et al, 2017).

As a common feature of ASD, impaired synaptic transmission has also been observed in UBE3A-dependent ASD. Previous research using *Ube3a* 2xTg mice has reported suppressed excitatory synaptic transmission, without a significant effect on the inhibitory synaptic function (Smith et al, 2011). However, the molecular mechanisms underlying the synaptic dysfunction in UBE3A-dependent ASD remain unclear. In this study, we report that the expression of AMPA receptors (AMPARs) and their synaptic distribution were significantly decreased in neurons with UBE3A overexpression and in *Ube3a* 2xTg brains. Interestingly, we found that the mRNA of the GluA1 subunit was restricted to the nucleus, causing a reduction in GluA1 protein synthesis. Further investigation revealed that a decrease in SARNP, an mRNA nuclear export factor, was responsible for the nuclear retention of GluA1 mRNA in UBE3A-overexpressing neurons. Indeed, restoration of SARNP levels successfully normalized mRNA distribution and rescued GluA1 expression, neuronal activity, and certain autistic behaviors in UBE3A-overexpressing conditions. These findings indicate that SARNP-mediated nuclear retention of GluA1 mRNA plays a crucial role in AMPAR expression and synaptic activity implicated in UBE3A-dependent ASD.

# Results

## AMPAR subunit GluA1 is decreased under UBE3A-overexpressing conditions, causing a reduction of AMPAR-mediated activity

It has been observed that *Ube3a* 2xTg ASD mice exhibit reduced excitatory synaptic transmission while maintaining intact inhibitory synaptic function (Smith et al, 2011). However, the mechanism underlying this impairment remains unclear. Because the ionotropic glutamate receptors are the primary mediators of most excitatory transmission in the brain, we wanted to examine if the expression of AMPA receptors (AMPARs) and NMDA receptors (NMDARs) was disrupted in *Ube3a* 2xTg brains (Fig. 1A). For this purpose, prefrontal cortical tissues were collected from the brains of postnatal day 21 (P21) wild-type (WT) and *Ube3a* 2xTg mice. Western blotting of brain lysates revealed a notable reduction in AMPAR subunit GluA1 in *Ube3a* 2xTg mice as compared to WT mice. In contrast, the expression of GluA2 remained unchanged (Fig. 1B). No significant alterations were observed in the expression levels of the NMDAR subunit NR1 and the GABA$_A$R subunit α1 in *Ube3a* 2xTg mice compared with their WT counterparts (Fig. 1B). We then examined the synaptic expression of GluA1 in day-in-vitro (DIV) 14 primary neurons, which were prepared from P0 WT or *Ube3a* 2xTg mice (Fig. 1C). The neurons were immunostained with VGluT and GluA1 as presynaptic and postsynaptic markers, respectively. A decrease in both GluA1 puncta and VGluT/GluA1 colocalized puncta was observed in *Ube3a* 2xTg neurons, indicating a reduced synaptic expression of GluA1 subunit (Fig. 1D).

To assess functional changes in AMPARs due to UBE3A overexpression, we analyzed AMPAR-mediated neuronal activation, as indicated by the expression of the immediate early gene c-Fos. Primary cultured neurons were infected with either a GFP virus as a control or a UBE3A virus to induce overexpression. C-Fos expression was examined at the basal condition or following glutamate stimulation, in the presence of TTX and APV to block action potentials and NMDARs, respectively (Fig. 1E). Glutamate treatment successfully increased the number of c-Fos positive cells in the GFP-infected control neurons, and this effect was blocked by the AMPAR antagonist CNQX, indicating that the enhancement of neuronal activity was mediated by AMPARs (Fig. 1F). However, glutamate only resulted in a minor increase in c-Fos in neurons with UBE3A overexpression, suggesting a reduction in AMPAR-mediated activity under UBE3A-overexpressing conditions (Fig. 1F).

## GluA1 mRNA is trapped in the nucleus of *Ube3a* 2xTg neurons

The reduction in GluA1 protein level could originate from various stages during GluA1 expression and maintenance, such as transcription, translation, and degradation. Given that UBE3A functions as a ubiquitin-protein E3 ligase and transcription co-activator, we investigated whether UBE3A targets GluA1 for

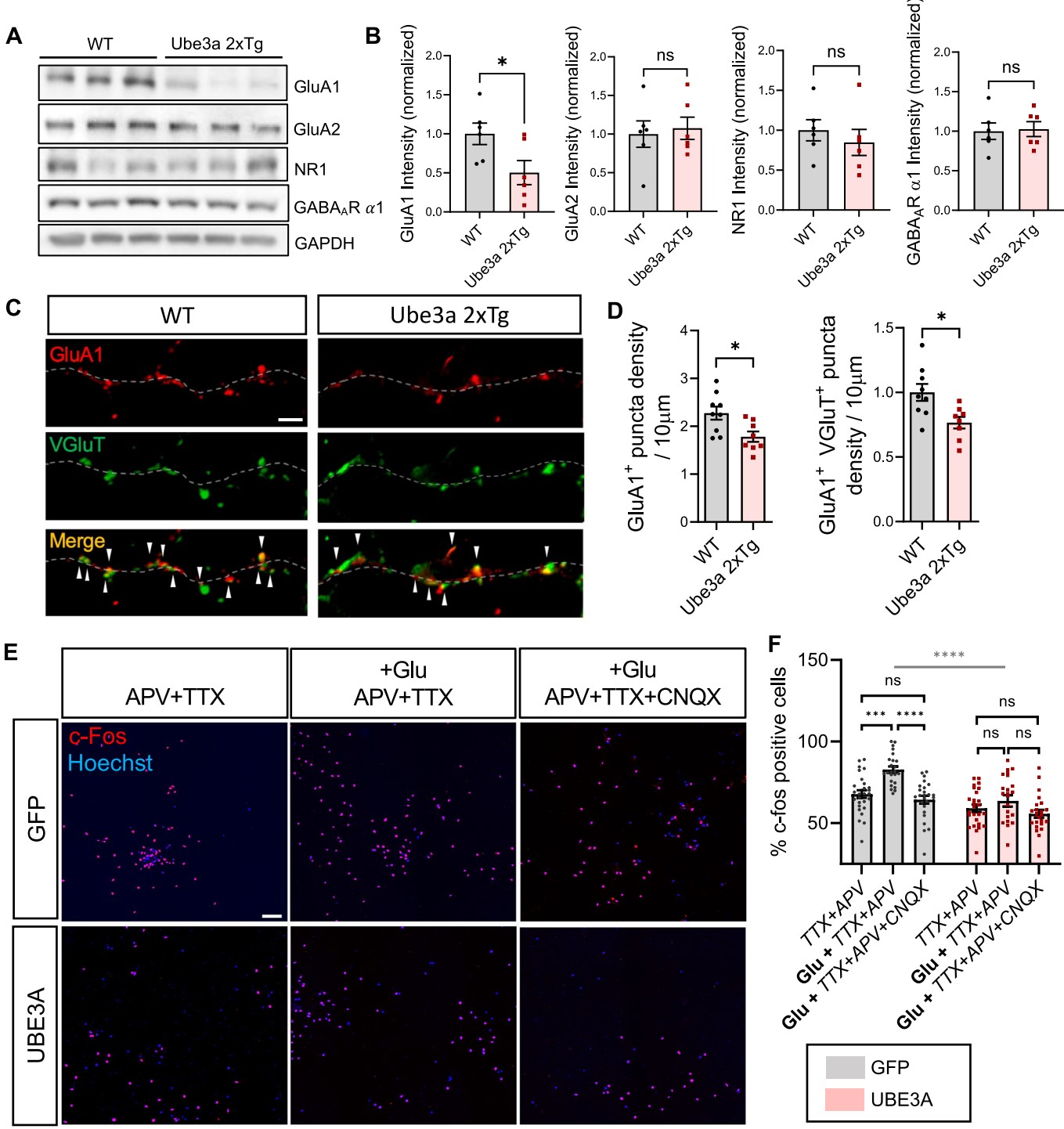

ubiquitination and degradation or suppresses GluA1 transcription. To determine the involvement of GluA1 ubiquitination, we transfected HEK293T cells with HA-ubiquitin (Ubi) and GluA1, along with either WT UBE3A or the E3 ligase dead mutant UBE3A C820A (Fig. 2A). Two days post-transfection, GluA1 was immunoprecipitated and probed for HA (ubi). We observed that GluA1 ubiquitination level in UBE3A-transfected neurons was comparable to that of the control group, and similar to the cells expressing the ligase dead UBE3A C820A (Fig. 2A). These findings

indicate that GluA1 is not a target protein of UBE3A for ubiquitination and degradation.

To determine the role of UBE3A in GluA1 transcription, we utilized fluorescent in situ hybridization (FISH) to label GluA1 mRNA in primary neurons (Fig. 2B). To validate the RNA FISH method, we probed GluA1 RNA in primary neurons that had been transfected with scramble (SCR)-GFP, GluA1 shRNA to knock-down GluA1, or a GluA1 construct to overexpress GluA1 (Fig. EV1A). As expected, we observed a reduction in the FISH intensity

**Figure 1.  UBE3A overexpression results in a reduction in GluA1 expression and AMPAR-mediated neuronal activity.**

(A) Western blotting of GluA1, GluA2, NR1, and GABA$_A$R α1 in the brain lysates of WT and *Ube3a* 2xTg mice. GluA1 expression was reduced in *Ube3a* 2xTg mice. (B) Quantitative analysis of GluA1, GluA2, NR1, and GABA$_A$R α1 in WT and *Ube3a* 2xTg mice. $n = 6$ animals/group. (C) Synaptic expression of GluA1 was identified by colocalized VGluT1 and GluA1 puncta fluorescent signals. Scale bars, 5 μm. (D) Density of GluA1 puncta in DIV14 primary neurons prepared from P0 WT or Tg mice. Quantification showed a decrease in density of GluA1 and GluA1/VGluT colocalized puncta in Tg neurons compared to WT neurons. $n = 9$ WT, $n = 8$ *Ube3a* 2xTg. (E) c-Fos expression in rat hippocampal neurons infected with either GFP or UBE3A virus. Neurons were treated with either DMSO or 10 μM Glu in the presence of APV + TTX or APV + TTX + CNQX. The nuclei were stained by Hoechst. Scale bars,100 μm. (F) Quantification of the percentage of c-Fos positive cells. Glutamate (Glu) treatment induced a significant increase of c-Fos positive neurons in GFP control but not UBE3A-overexpressing condition, which was blocked by AMPAR antagonist CNQX, indicating that the increase was mediated by AMPARs. GFP-TTX + APV: $n = 27$, GFP-Glu+TTX + APV: $n = 24$, GFP-Glu+TTX + APV + CNQX: $n = 25$, GFP + UBE3A-TTX + APV: $n = 26$, GFP + UBE3A-Glu+TTX + APV: $n = 23$, GFP + UBE3A-Glu+TTX + APV + CNQX: $n = 23$ images of 2.25 mm$^2$ area. Data information: in (B, D), unpaired two-tailed $t$ test. In (F), two-way ANOVA with Bonferroni's multiple comparisons test. Mean ± SEM. *$p < 0.05$; ***$p < 0.001$; ****$p < 0.0001$; ns = not significant.

of GluA1 mRNA when GluA1 was knocked down by shRNA, and an enhancement when GluA1 was overexpressed (Fig. EV1B), which primarily affected the cytosolic GluA1 mRNA (Fig. EV1C). In primary neurons prepared from P0 WT and *Ube3a* 2xTg mice, we found that the total fluorescence signal of GluA1 mRNA in the soma was comparable between WT and *Ube3a* 2xTg neurons (Fig. 2C). However, when the cytosolic and nuclear GluA1 mRNA levels were separately measured, we observed a significant decrease in cytosolic GluA1 mRNA and a corresponding increase in nuclear GluA1 mRNA in *Ube3a* 2xTg neurons (Fig. 2D). We then fractionated brain lysates into nuclear and cytosolic compartments, and analyzed the isolated RNAs from each fraction by real-time PCR (qPCR). Similar results were obtained showing that GluA1 mRNA levels were reduced in the cytosol and increased in the nucleus of *Ube3a* 2xTg brains (Fig. 2F), while the total GluA1 mRNA levels remained comparable between the two groups (Fig. 2E). In contrast, the mRNA expression and nuclear distribution of GluA2, the other major subunit of AMPAR, were unaffected by UBE3A overexpression (Fig. EV2A–C). This is consistent with the intact GluA2 protein expression observed in *Ube3a* 2xTg brains (Fig. 1A,B). These findings suggest that GluA1 mRNA is trapped in the nucleus of neurons with a high dosage of *Ube3a*, potentially contributing to the observed reduction of GluA1 protein.

## Downregulation of mRNA nuclear export gene *SARNP* leads to GluA1 mRNA nuclear retention in UBE3A-overexpressing neurons

To investigate potential candidates implicated in the nuclear retention phenotype of GluA1 mRNA, we collected prefrontal cortical tissue from P5 WT and *Ube3a* 2xTg mice for RNA sequencing. Analysis of the RNAseq dataset identified several dysregulated mRNA nuclear transport genes in *Ube3a* 2xTg animals (Fig. 3A). *Sarnp*, also known as *Cip29*, showed the most significant alteration, exhibiting an average 55% decrease in RNA expression in *Ube3a* 2xTg samples. SARNP protein interacts with UAP56, an RNA helicase component of the transcription-export (TREX) complex that mediates mRNA splicing and export (Kang et al, 2020; Dufu et al, 2010; Schumann et al, 2016). Prior research has shown that SARNP knockdown is able to suppress mRNA export (Kang et al, 2020), therefore its downregulation in *Ube3a* 2xTg animals could be implicated in GluA1 mRNA mislocalization. Hhex, the second top candidate involved in mRNA nuclear transport (Fig. 3A), has been reported to inhibit the eIF4E-dependent mRNA nuclear export and subsequent translation

(Topisirovic et al, 2003). However, downregulation of *Hhex*, as seen in *Ube3a* 2xTg animals, would be expected to enhance mRNA export, which is opposite to the observed change of GluA1 mRNA in *Ube3a* 2xTg neurons. The third candidate, *Eny2*, is associated with both the TREX-2 and SAGA complexes and is involved in mRNA export coupled with transcription activation (Jani et al, 2012). Therefore, the decreased expression of *Sarnp* and *Eny2*, but not *Hhex*, may potentially underlie the GluA1 mRNA nuclear retention. Further analysis of these targets at the protein levels confirmed a reduction of SARNP protein, as verified by immunocytochemistry in DIV8 primary neurons prepared from P0 WT and 2xTg mice (Fig. 3B,C), and by western blotting of P21 WT and 2xTg prefrontal cortical lysates (Fig. 3D,E). On the contrary, ENY2 protein levels were only slightly reduced in P21 2xTg prefrontal cortical lysates compared to WT (Fig. 3F,G). To assess the roles of SARNP and ENY2 in 2xTg neurons, we replenished them into primary neurons prepared from P0 *Ube3a* 2xTg mice (Fig. 3H). Quantification of RNA FISH fluorescence intensity revealed that SARNP transfection rescued the localization of GluA1 mRNA back to WT levels, whereas ENY2 transfection failed to rescue the phenotype and instead led to a decrease in GluA1 mRNA levels (Fig. 3I). A similar rescue effect of SARNP was also observed in rat primary neurons transfected with UBE3A (Appendix Figure S1). HHEX protein levels in P21 2xTg prefrontal cortical lysates also showed a significant decrease compared to WT (Fig. EV3A,B). Yet, when HHEX was replenished in UBE3A-overexpressing neurons, it did not rescue the observed nuclear retention of GluA1 mRNA as shown by RNA FISH (Fig. EV3C,D). Therefore, to further characterize the role of SARNP in GluA1 mRNA nuclear localization, we knocked down SARNP in WT rat hippocampal neurons using shRNAs, resulting in an accumulation of GluA1 mRNA in the nucleus, without altering the total level of GluA1 mRNA in the soma (Fig. 3J–L), which mimicked the phenotype observed in UBE3A-overexpressing neurons. The efficacy of the SARNP shRNAs was validated in both HEK cells and primary neurons (Fig. EV4A–C). These findings indicate that the UBE3A-caused downregulation of SARNP is responsible for GluA1 mRNA nuclear retention.

## SARNP restoration rescues UBE3A-dependent reduction in GluA1 protein synthesis

Suppression of mRNA nuclear export is expected to reduce mRNA availability for protein translation in the cytoplasm. Therefore, we hypothesized that the synthesis rate of GluA1 protein is compromised under conditions of UBE3A overexpression. To test

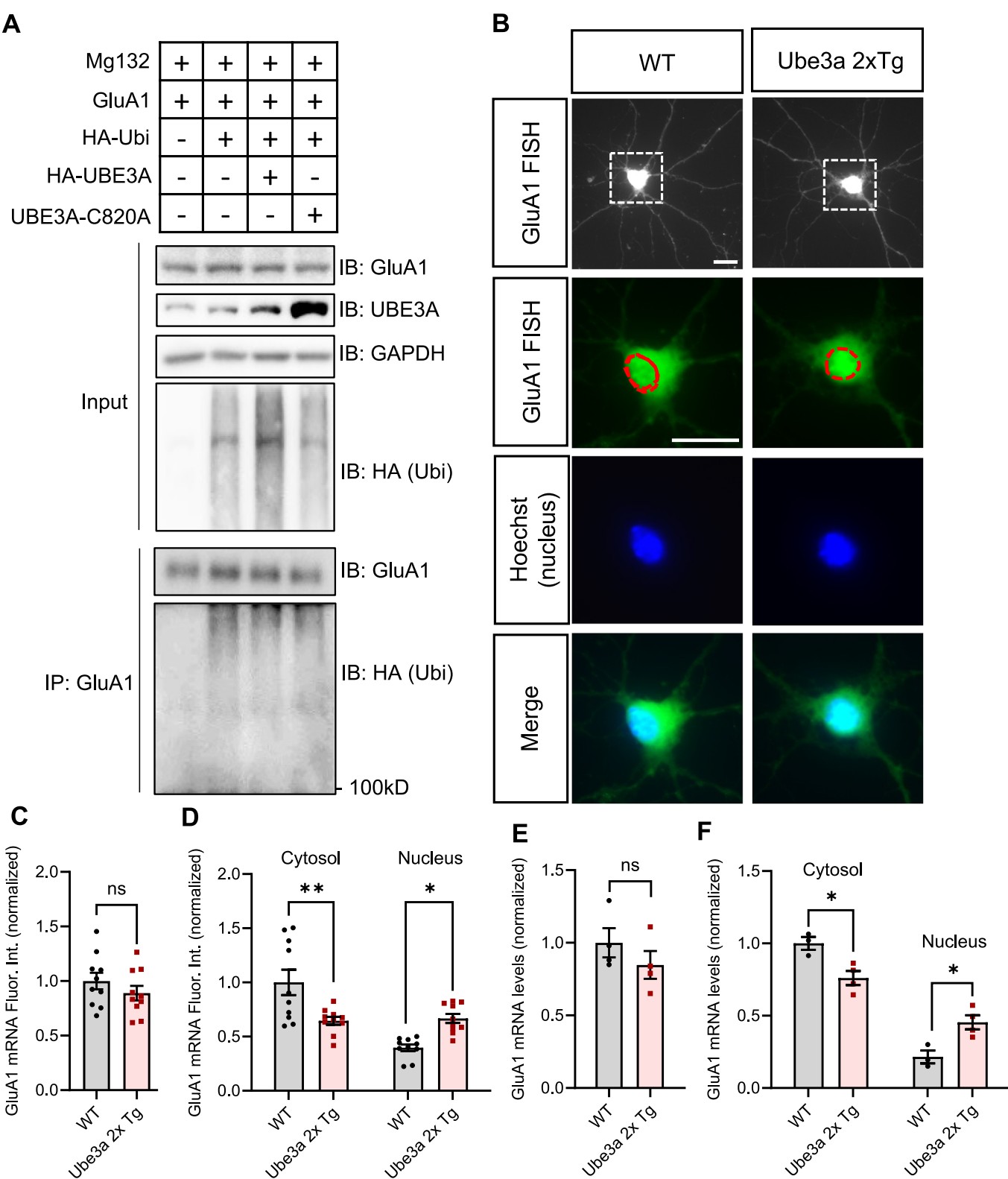

this hypothesis, we employed puromycin assays to label and identify the newly synthesized GluA1 peptides. As a structural analog of tyrosyl-tRNA, puromycin can be specifically incorporated into and label nascent polypeptide chains (Schmidt et al, 2009). DIV15 cortical primary neurons were incubated with 1 μM

puromycin for 12 h to label newly synthesized peptides, after which GluA1 peptides were isolated by immunoprecipitation (IP) using antibodies against the N-terminus of GluA1 and probed for puromycin (Fig. 4A). The quantification revealed a significant decrease in the level of puromycin-tagged GluA1 peptides in

◄  **Figure 2.   GluA1 mRNA is accumulated in the nucleus of *Ube3a* 2xTg neurons.**

(A) GluA1 ubiquitination assay. HEK293 cells were transfected with GluA1 vector and HA-ubiquitin (ubi), together with either a control vector, UBE3A, or the E3 ligase-dead mutant UBE3A C820A for 2 days. GluA1 was immunoprecipitated and probed for ubi. Cell lysates (input) were also probed to detect total protein levels. (B) GluA1 RNA FISH in DIV10 primary neurons prepared from P0 WT and Tg mice. Scale bars, 15 µm. The dotted circle highlights the nucleus location, which was stained using Hoechst. (C) Total intensity of GluA1 mRNA fluorescence was not changed in Tg neurons. $n = 10$ neurons/group. (D) Subcellular distribution of GluA1 mRNA. Cytosolic GluA1 mRNA was reduced, whereas the nuclear GluA1 mRNA was increased in Tg neurons. $n = 10$ neurons/group. (E) qPCR of total GluA1 mRNA in whole brain lysates showed no difference between WT and Tg mice. $n = 4$ animals/group (F) GluA1 mRNA expression was decreased in the cytosolic fraction and increased in the nuclear fraction of brain lysates from Tg mice. $n = 3$ WT; $n = 4$ *Ube3a* 2xTg. Data information: Mean ± SEM. *$p < 0.05$; **$p < 0.01$; ns = not significant. In (D, F), two-way ANOVA with Bonferroni's multiple comparisons test. In (C, E), unpaired two-tailed *t* test.

UBE3A-overexpressing neurons (Fig. 4B). Importantly, this decrease was reversed by SARNP restoration, indicating that SARNP-caused GluA1 mRNA nuclear retention contributes to the suppressed GluA1 protein synthesis. Consistently, in neurons infected with the UBE3A virus, the reduction of GluA1 protein abundance was also rescued by SARNP replenishment (Fig. 4C), indicating that suppressed GluA1 protein synthesis caused by SARNP downregulation leads to a reduction in GluA1 protein levels. In contrast, the protein synthesis rate and protein abundance of GluA2 remained unaltered when UBE3A was overexpressed (Fig. EV2D), suggesting that the UBE3A effect is specific to GluA1 subunit.

To further evaluate whether the SARNP-mediated pathway is responsible for synaptic GluA1 reduction, primary cortical neurons were transfected with either GFP, GFP + UBE3A or GFP + UBE3A + SARNP. Double immunostaining for GluA1 and PSD95 was performed to visualize the GluA1 expression at postsynaptic sites (Fig. 4D). As expected, UBE3A overexpression significantly reduced GluA1 immunointensity at synapses and the soma. Importantly, these changes were successfully reversed by overexpression of SARNP (Fig. 4E–G), highlighting a crucial role for SARNP-mediated GluA1 mRNA nuclear export in UBE3A-induced alterations in GluA1 expression.

## SARNP restoration rescues AMPAR-mediated synaptic currents in UBE3A-overexpressing conditions

Given that GluA1 is a major subunit of AMPARs, we sought to determine whether the SARNP-mediated GluA1 dysregulation is responsible for the impaired AMPAR activity observed in UBE3A-overexpressing conditions (Fig. 1E,F). To this end, we first evaluated neuronal activation-induced c-Fos expression in primary cultured neurons when AMPARs were selectively activated. As anticipated, glutamate incubation in the presence of APV and TTX (to block NMDARs and action potentials respectively) significantly elevated the percentage of c-Fos positive cells in control neurons infected with GFP virus, and this effect was abolished by UBE3A overexpression (Fig. 5A,B). Importantly, the introduction of SARNP together with UBE3A viruses successfully rescued this phenotype and allowed glutamate treatment to enhance c-Fos expression (Fig. 5B). Of note, in both GFP-infected control neurons and neurons co-infected with UBE3A and SARNP viruses, the increase in c-Fos positive cells was suppressed by AMPAR antagonist CNQX, indicating that the increases in activity were mediated by AMPARs (Fig. 5B). Furthermore, we assessed AMPAR-mediated currents by patch-clamp recording of the miniature excitatory postsynaptic currents (mEPSCs). Primary cortical neurons were infected with a virus expressing either GFP or

UBE3A. Additionally, these neurons were co-infected with either a SARNP or GluA1 virus for potential rescue (Fig. 5C). Patch-clamp recordings revealed that UBE3A overexpression significantly decreased the amplitude of AMPAR mEPSCs as compared to control neurons (Fig. 5D), while the frequency and the decay time constant of AMPAR mEPSCs were not affected (Fig. 5E,F). As expected, viral expression of GluA1 abolished UBE3A-overexpression induced reduction in mEPSC amplitude (Fig. 5D). Importantly, viral infection of SARNP successfully rescued the amplitude of AMPAR mEPSCs, without affecting their frequency and decay kinetics (Fig. 5D–F). These findings support a molecular cascade where the downregulation of SARNP, causing the suppression of GluA1 mRNA nuclear export and GluA1 synthesis, leads to weakened synaptic function in UBE3A-overexpressing conditions.

## Restoring SARNP rescues neuronal activity and autistic behaviors in *Ube3a* 2xTg mice

Previous studies suggest that defective AMPAR-mediated excitatory synaptic transmission is associated with autistic behaviors (Etherton et al, 2011; Wiedholz et al, 2007; Kim et al, 2018). AMPAR dysregulation has been observed in numerous ASD animal models and in the post-mortem brains of human patients with autism (Purcell et al, 2001). To investigate the role of SARNP-mediated AMPAR dysfunction in the development of autistic phenotypes in vivo, we examined whether replenishing SARNP could rescue autistic phenotypes in *Ube3a* 2xTg mice. For this purpose, we performed bilateral intracerebroventricular (ICV) injections of the GFP-SARNP lentivirus (LV-GFP-SARNP) at P0 to restore SARNP levels in *Ube3a* 2xTg brains (2 µl each side). The WT and 2xTg animals injected with the GFP virus (LV-GFP) served as controls (Fig. 6A). Behavioral tests were conducted between P30 to P40, and the brains of the infected animals were then processed for biochemical analysis at around P40 (Fig. 6A). Imaging of brain slices and western blotting of GFP-SARNP in the infected brains confirmed the viral expression throughout the brain (Figs. 6B and EV5A,B). Firstly, to examine the impact of SARNP replenishment on neuronal activity, we performed immunohistochemistry (IHC) to probe c-Fos expression in the prefrontal cortex (PFC) and hippocampus (HPC) (Fig. 6C). We found that c-Fos expression was decreased in the PFC of *Ube3a* 2xTg brains, and this reduction was partially rescued by SARNP viral infection (Fig. 6D). We then analyzed the medial prefrontal cortex (mPFC) that includes the prelimbic cortex, infralimbic cortex and the anterior cingulate cortex, a critical region related to social behaviors (Ko, 2017; Lee et al, 2016; Sun et al, 2020). We observed a decline in c-Fos expression in *Ube3a* 2xTg mice, which was

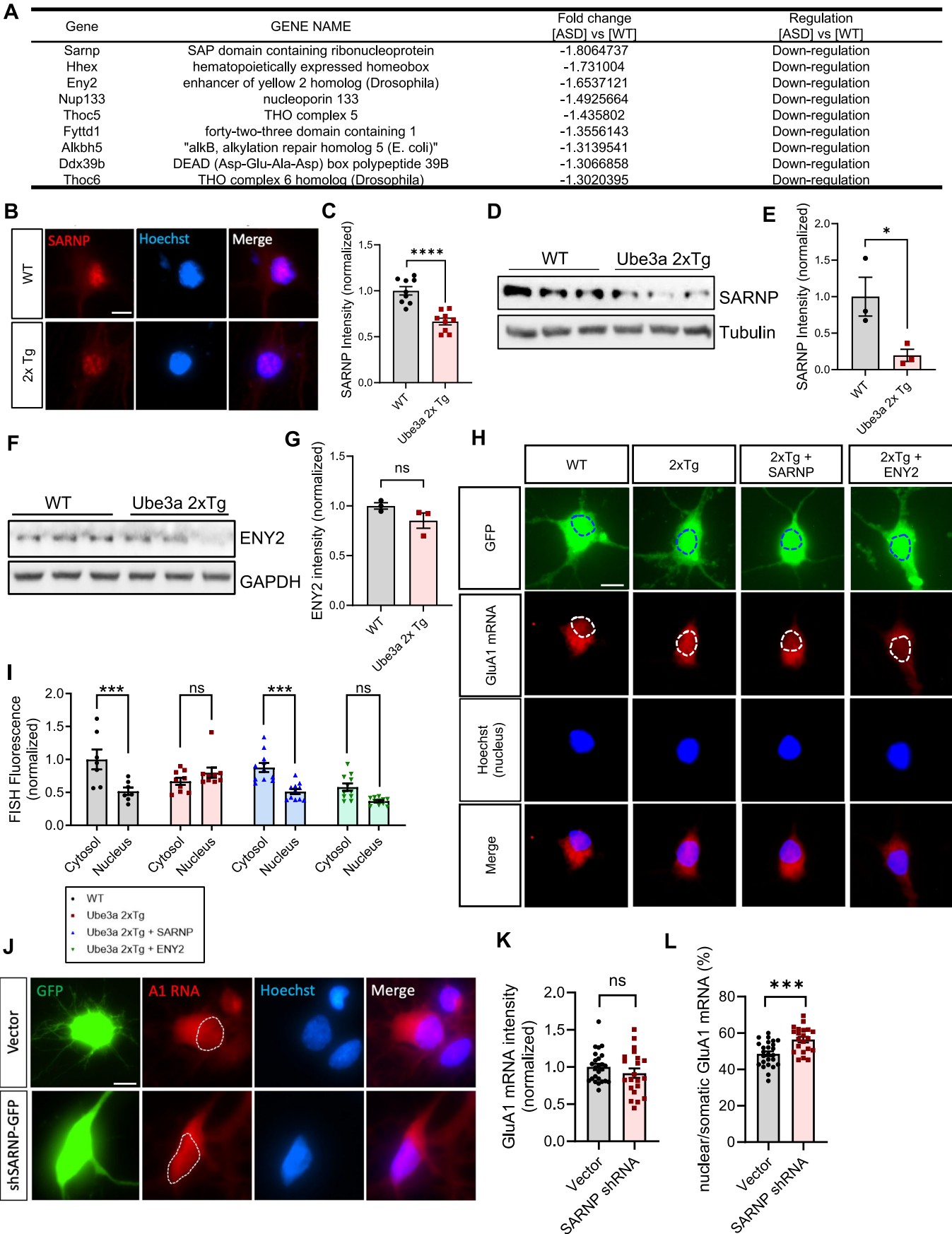

◄  **Figure 3.    Downregulation of the mRNA nuclear transport gene *SARNP* causes GluA1 mRNA nuclear retention.**

(A) Top dysregulated candidates mediating mRNA nuclear transport in *Ube3a* 2xTg brains identified by RNA-seq. (B, C) Immunostaining showed that SARNP expression was reduced in *Ube3a* 2xTg primary mouse neurons. n = 9 neurons/group. Scale bars, 10 μm. (D) Protein expression of SARNP in the brain lysates of WT and *Ube3a* 2xTg mice was analyzed by western blot. (E) Quantitative analysis showed a significant reduction of SARNP in *Ube3a* 2xTg brains. n = 3 animals/group. (F) Protein expression of ENY2 in the brain lysates of WT and *Ube3a* 2xTg mice was analyzed by western blot. (G) Quantitative analysis of ENY2 in *Ube3a* 2xTg brains. n = 3 animals/group. (H) WT and *Ube3a* 2xTg mouse cultures were transfected with GFP, GFP + SARNP, or GFP + ENY2 at DIV10. GluA1 mRNA (red) was probed by RNA FISH at DIV14. The dotted circle indicates the location of nucleus. Hoechst was used to stain the nucleus. Scale bars, 15 μm. (I) Quantification of GluA1 mRNA distribution showed that SARNP restoration was able to rescue GluA1 mRNA nuclear retention. n = 7 WT, n = 9 *Ube3a* 2xTg, n = 11 *Ube3a* 2xTg+SARNP, n = 11 *Ube3a* 2xTg + ENY2. (J) Primary rat neurons were transfected with either a control vector, or shSARNP-GFP to knock down SARNP at DIV7. Neurons were fixed at DIV10 and processed for GluA1 RNA FISH. The dotted circle indicates the location of nucleus. Hoechst was used to stain the nucleus. Scale bars, 10 μm. (K) Quantification of total GluA1 mRNA intensity in the soma. SCR vector: n = 25, shSARNP-GFP: n = 21. (L) Quantification of GluA1 mRNA intensity in the nucleus vs. in the soma. SCR vector: n = 25, shSARNP-GFP: n = 21. Data information: Mean ± SEM. *p < 0.05; ***p < 0.001; ****p < 0.0001; ns = not significant. In (C, E, G, K, L), unpaired two-tailed t test. In (I), two-way ANOVA with Bonferroni's multiple comparisons test.

partially restored by LV-GFP-SARNP (Fig. 6E). In the HPC, the percentage of c-Fos positive cells was reduced in *Ube3a* 2xTg mice as well, which was reversed to WT levels by ICV injection of LV-GFP-SARNP (Fig. 6F). These findings indicate that SARNP insufficiency is responsible for the aberrant neuronal activity in *Ube3a* 2xTg mice.

We then wanted to know if the reduction in SARNP also contributes to the development of autistic behaviors. We first investigated whether restoring SARNP by ICV injection of LV-GFP-SARNP at P0 could rescue impaired social communication in *Ube3a* 2xTg mice. To this end, we recorded and analyzed their ultrasonic vocalizations (USVs). Pups isolated from the mother and littermates at P9 were placed in a recording chamber for 5 min in which a microphone recorded their vocalization signals (Fig. 7A). In *Ube3a* 2xTg mice with LV-GFP injection, we observed an increase in the number of calls and total call duration compared to WT males (Fig. 7B,C), which is a common feature observed in ASD mouse models (Premoli et al, 2021, 2019; Schmeisser et al, 2012; Lai et al, 2014; Scattoni et al, 2011). However, SARNP restoration failed to reverse these changes (Fig. 7B,C). Furthermore, *Ube3a* 2xTg animals injected with LV-GFP virus did not show significant changes in their mean call duration, peak frequency of calls, and peak amplitude of calls, and LV-GFP-SARNP injection showed little effect on these features (Fig. 7D–F). These findings suggest that *Ube3a* over-dosage results in impaired social communication in *Ube3a* 2xTg animals; however, this phenotype is independent from the SARNP-mediated pathway.

Impairments in social interaction are a hallmark of ASD. To determine if SARNP-mediated alterations contribute to impaired sociability in *Ube3a* 2xTg mice, we performed the three-chamber sociability test. For the social preference test, a stranger mouse was placed into the cage at either side chambers, and the test mouse of the same sex was allowed to explore the apparatus freely (Fig. 7G). As expected, impaired social preference was found in *Ube3a* 2xTg mice injected with LV-GFP (Fig. 7I). Importantly, LV-GFP-SARNP injection successfully restored the preference for stranger mice in *Ube3a* 2xTg mice (Fig. 7I). For the social novelty test, a second mouse of the same sex (novel mouse) was introduced into the remaining empty chamber, and the test mouse was allowed to interact with both mice (Fig. 7H). LV-GFP-SARNP was also effective in rescuing the abnormal preference for the novel mouse observed in *Ube3a* 2xTg animals (Fig. 7J). These results indicate a crucial involvement of SARNP in the social abnormalities in ASD.

Repetitive behavior is another core phenotype in individuals with ASD, we therefore examined the repetitive self-grooming behavior in *Ube3a* 2xTg mice. We noted that *Ube3a* 2xTg mice injected with LV-GFP virus displayed longer grooming time compared to their WT counterparts (Fig. 7K), but in 2xTg mice injected with LV-GFP-SARNP, the grooming behavior was similar to that of the WT mice (Fig. 7K).

Taken together, our results demonstrate that restoring SARNP in the brain at P0 is able to rescue several core autistic phenotypes in *Ube3a* 2xTg animals, including neuronal activity, social interaction, and grooming behaviors. These findings underline the significance of the SARNP-regulated AMPAR expression and synaptic activity in the expression of autistic phenotypes in UBE3A-dependent ASD.

## Discussion

As one of the most prevalent ASD-causing genes, *UBE3A* over-dosage and hyperactivity have been implicated in ASD patients (Yi et al, 2015; Iossifov et al, 2014; Weston et al, 2021; Finucane et al, 2021; Khatri and Man, 2019; Smith et al, 2011; Noor et al, 2015). Previous studies on *Ube3a* ASD mice have revealed a weakened strength in excitatory synaptic transmission (Smith et al, 2011). However, molecular abnormalities in synapses, the underlying mechanisms, and their relevance to the autistic phenotype remain unclear. In this study, we show that UBE3A overexpression traps GluA1 mRNA in the nucleus of neurons due to reduced expression of the mRNA export protein SARNP. This change in mRNA trafficking results in a suppression of GluA1 protein synthesis, and subsequently, a decrease in GluA1 protein abundance and AMPAR-mediated synaptic function. Notably, SARNP restoration successfully rescued mRNA localization and protein expression of GluA1, leading to a normalization in AMPAR-mediated currents, neuronal activity, and autistic behaviors in *Ube3a* 2xTg mice. These findings highlight the vital role of SARNP-mediated GluA1 mRNA nuclear transport in UBE3A-dependent ASD.

AMPARs, as the primary mediator for excitatory synaptic transmission, have been previously reported to be dysregulated at various levels in ASD, including altered gene expression, alternative splicing, RNA modification, receptor trafficking, and subunit composition (Li et al, 2016a; Achuta et al, 2018; Yennawar et al, 2019; Niescier and Lin, 2021; Tramarin et al, 2018). Our study reveals a novel mechanism in AMPAR dysregulation in UBE3A-dependent ASD, i.e., defective nuclear-cytosol trafficking of

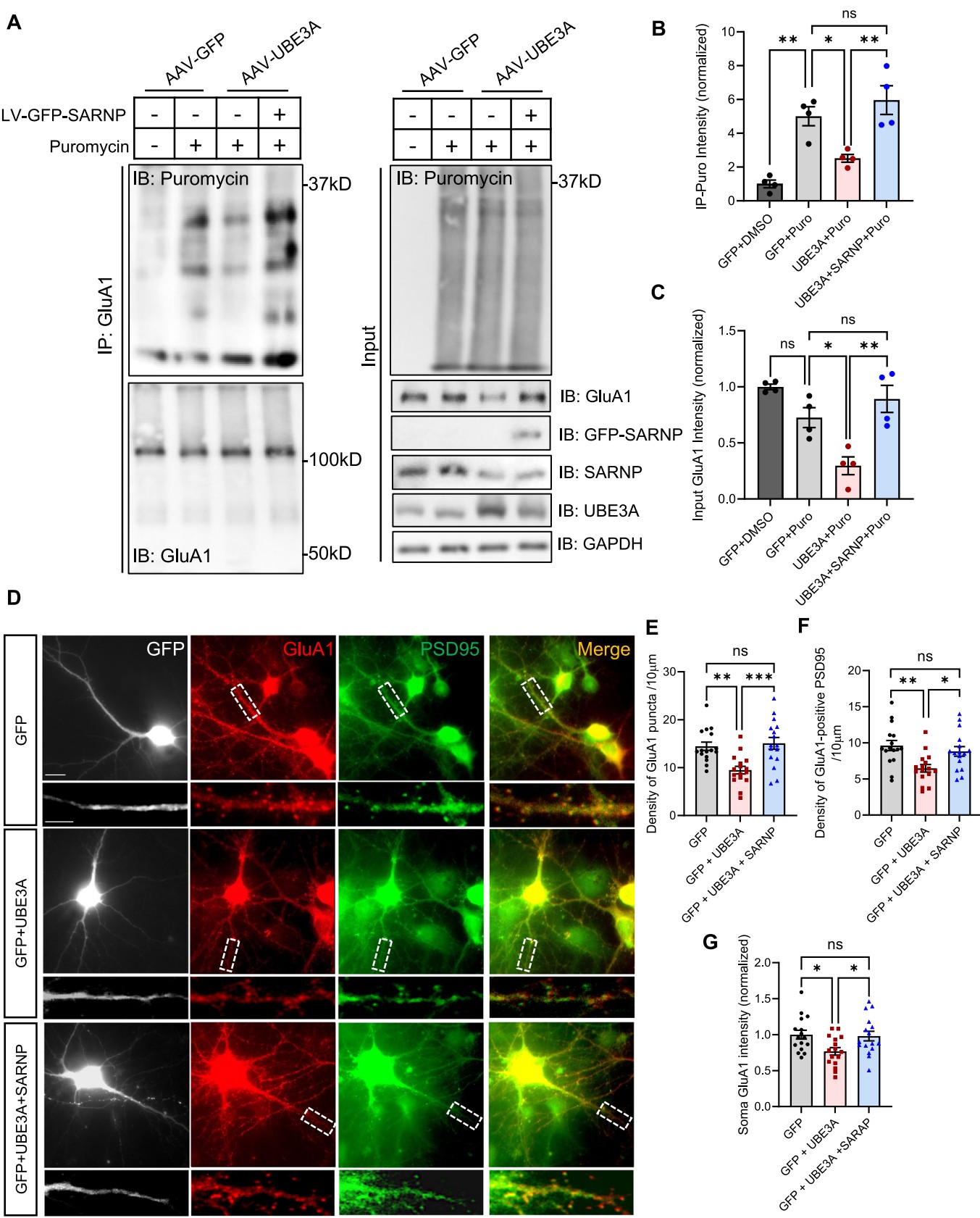

◀ **Figure 4. UBE3A overexpression reduces GluA1 protein synthesis and abundance, which are rescued by SARNP restoration.**

(A) Primary rat neurons were infected with either GFP virus, UBE3A virus, or both UBE3A and SARNP viruses. Neurons were treated with DMSO or puromycin for 12 h at DIV 11. After that, GluA1 was immunoprecipitated and probed for puromycin to detect GluA1 peptides being synthesized. Cell lysates (input) were also probed to detect total protein levels. (B) Quantification of puromycin-labeled GluA1 showed a decrease in GluA1 protein synthesis with UBE3A overexpression, which was reversed by SARNP restoration. $n = 4$ independent experiments. (C) Quantification of GluA1 abundance in input lysates. UBE3A overexpression caused a decrease in GluA1 abundance in total lysates, which was rescued by SARNP restoration. $n = 4$ independent experiments. (D) DIV10 rat primary neurons were transfected with GFP, GFP + UBE3A, or GFP + UBE3A + SARNP. GluA1 (red) and PSD95 (green) were immunostained to measure somatic and synaptic expression of GluA1 at DIV14. Scale bars, 25 µm (full picture); 10 µm (enlarged area). (E–G) SARNP restoration rescued the density of GluA1 puncta (E) and colocalized GluA1 + PSD95 puncta (F) along the dendrites, and GluA1 intensity in the soma (G). GFP: $n = 16$, GFP + UBE3A: $n = 16$, GFP + UBE3A + SARNP: $n = 16$. Data information: Mean ± SEM. *$p < 0.05$; **$p < 0.01$; ***$p < 0.001$; ns = not significant. One-way ANOVA with Bonferroni's multiple comparisons test.

AMPAR mRNA leading to a suppression in AMPAR synthesis and synaptic accumulation. Specifically, abnormal translation of the AMPAR subunit, resulting from changes in expression or alternative splicing, has been implicated in the pathophysiology and neurological phenotype of both fragile X syndrome and Rett syndrome (Achuta et al, 2018; Li et al, 2016a; Tramarin et al, 2018). Aberrant AMPAR trafficking can lead to an imbalanced E/I distribution in individual neurons and neural circuits, which is proposed to be a key cellular target of ASD pathology (Nelson and Valakh, 2015; Lee et al, 2017; Sacai et al, 2020). Mutations in various ASD-related genes, such as *FMRP* in Fragile X syndrome and *MECP2* in Rett syndrome, have been linked to AMPAR trafficking defects (Nakamoto et al, 2007; Li et al, 2016b). Similarly, other autism candidate genes including *NLGNs, IQSEC2, DOCK4*, and *STXBP5*, are also implicated in AMPAR trafficking (Niescier and Lin, 2021). Altered AMPAR subunit composition, which can be induced by several ASD genes such as *SHANK2* and *SHANK3*, could impair synaptic maturation and circuit formation (Ha et al, 2018). Additionally, it has been shown that AMPAR modulation can normalize impaired excitatory transmission and social impairments in animal models of ASD (Kim et al, 2018). Thus, AMPAR regulation is one of the most common molecular substrates in ASD pathogenesis.

Previous research showed that the *Ube3a* 2xTg mouse model displayed a decrease in both the amplitude and frequency of mEPSCs (Smith et al, 2011), while in our study, we observed a reduction in amplitude without marked changes in the frequency of mEPSCs. This discrepancy could potentially originate from the use of different preparations. Smith et al recorded layer 2/3 neurons of the barrel cortex in brain slices from *Ube3a* 2xTg animals (Smith et al, 2011), whereas we utilized UBE3A-overexpressing cultured cortical neurons. The in vivo neural circuits and synaptic connectivity should significantly differ from the circumstances of primary neurons in vitro.

Under conditions of UBE3A overexpression, we detected a decrease in SARNP protein, both in vitro and in vivo. Because SARNP is not a target for UBE3A-induced ubiquitination, the decrease unlikely results from protein degradation (Fig. 2A). In fact, in addition to functioning as an E3 ubiquitin ligase, UBE3A also serves as a transcription coactivator, suggesting an involvement of transcriptional regulation. In line with this, our RNA sequencing analysis of *Ube3a* 2xTg brains demonstrated a reduction in SARNP mRNA level (Fig. 3A). Transcription factors can have dual functions, either activating or repressing the expression of target genes in different contexts (Latchman, 2001; Boyle and Després, 2010). It is possible that UBE3A activates the function or expression of a transcription factor which then

suppresses SARNP expression. Moreover, previous research has shown that transcriptional coactivators could also suppress gene transcription under different contexts (Shin et al, 2020; Ma et al, 2021; Nikolenko et al, 2005).

Our results confirm a crucial role for SARNP in UBE3A-dependent GluA1 downregulation. SARNP is a novel mRNA export protein. It interacts with UAP56, an RNA helicase component of the TREX complex, which facilitates mRNA splicing and export. Our findings suggest that dysregulation of SARNP contributes to the nuclear retention of GluA1 mRNA. However, the exact role of SARNP in the regulation of GluA1 mRNA export remains unclear. One possibility is that SARNP functions as a transport receptor, directly interacting with target mRNA and nucleoporins to facilitate mRNA export (Katahira, 2012). Alternatively, SARNP could assist in loading associated factors onto GluA1 mRNA to form functional mRNA-protein complexes (mRNPs) for nuclear export (Katahira, 2012). Another possibility is that SARNP might be involved in mRNA splicing, which is functionally coupled with mRNA nuclear export and promotes rapid and efficient mRNA export in mammalian cells (Valencia et al, 2008; Masuda et al, 2005; Luo and Reed, 1999). Typically, properly spliced mRNAs are predominantly found in the cytoplasm, whereas unspliced or mis-spliced pre-mRNAs are primarily retained in the nucleus (Valencia et al, 2008; Luo and Reed, 1999). In addition, the regulation on AMPARs seems to be GluA1 specific. In contrast to GluA1, we found that GluA2 protein expression was not altered in *Ube3a* 2xTg mice (Fig. 1A,B), and no significant changes in GluA2 mRNA expression or nuclear localization were observed in UBE3A-overexpressing neurons (Fig. EV2A–C). Moreover, the synthesis rate of GluA2 peptides remained stable, as demonstrated by the puromycin assay (Fig. EV2D). These findings suggest that GluA2 mRNA is not governed by the SARNP-mediated nuclear export machinery. Further studies will help to elucidate the detailed steps by which SARNP regulates AMPAR mRNA nuclear trafficking.

In *Ube3a* 2xTg ASD mice, social interaction and repetitive self-grooming behaviors, but not social communication, were restored following ICV injection of LV-GFP-SARNP virus at P0. The lack of rescue effect on social communication may suggest other cellular and molecular events in mediating this behavior. Alternatively, it could be due to insufficient LV-GFP-SARNP viral expression in the brain, as the ultrasonic vocalization recordings of pup isolation calls were performed 9 days after viral injection (P9), which was a shorter timeframe compared to the other behavioral tests performed at P30-P40. Isolation-induced ultrasonic vocalizations in neonates are regulated by the hypothalamus (Zimmer et al, 2019). It's possible that the LV-GFP-SARNP virus failed to induce

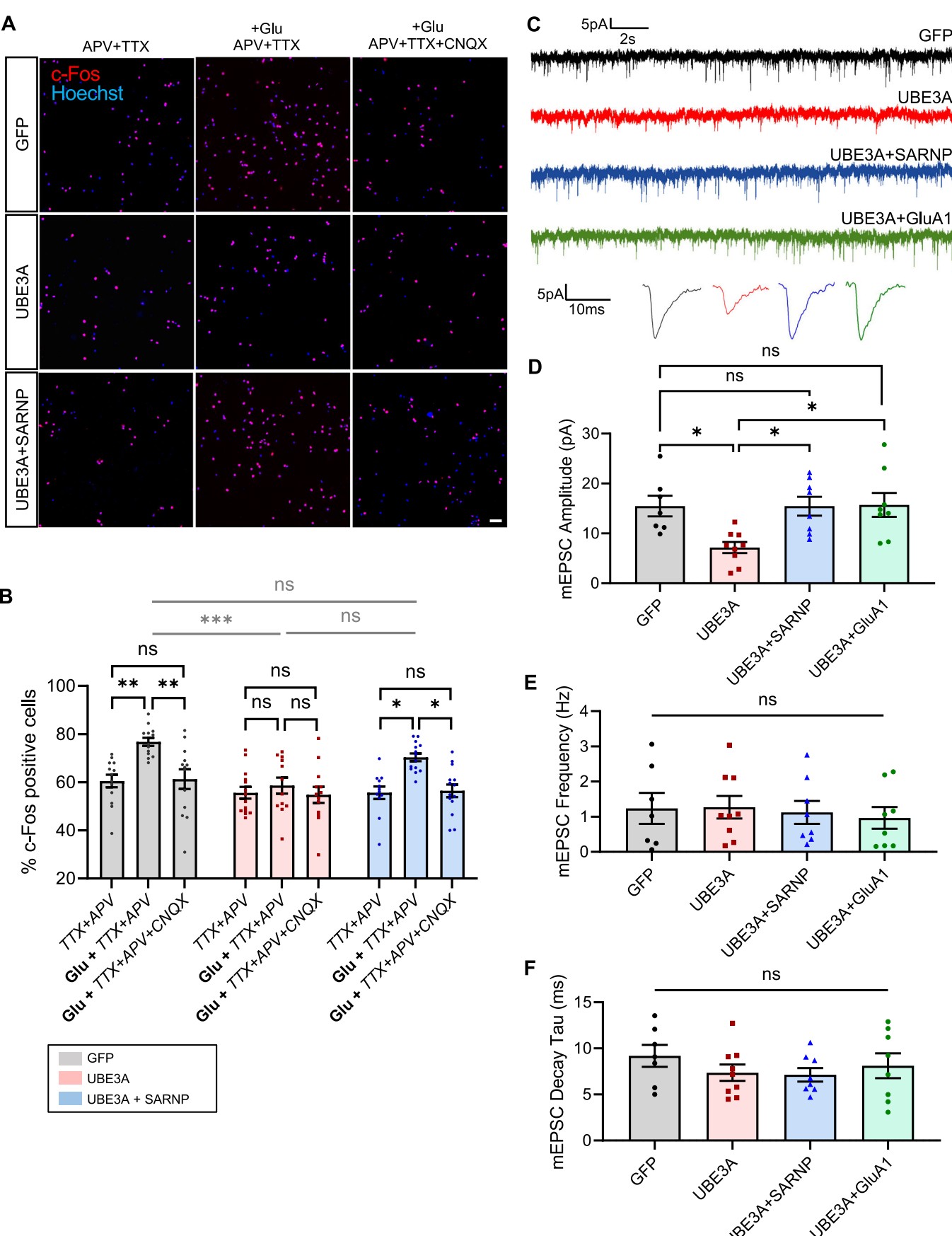

Figure 5.   UBE3A overexpression impairs AMPAR-mediated neuronal activity, which is rescued by SARNP restoration.

(A) c-Fos expression in rat primary neurons infected with GFP virus, UBE3A virus, or both UBE3A and SARNP viruses. Neurons were treated with either DMSO or 10 µM Glutamate (Glu) in the presence of APV + TTX or APV + TTX + CNQX. Hoechst was used to stain the nucleus. Scale bars,100 µm. (B) Quantification of the percentage of c-Fos positive cells. UBE3A overexpression abolished the Glu-induced increase in c-Fos positive neurons, which was rescued by SARNP restoration. GFP-TTX + APV: $n = 13$, GFP-Glu+TTX + APV: $n = 13$, GFP-Glu+TTX + APV + CNQX: $n = 13$, GFP + UBE3A-TTX + APV: $n = 14$, GFP + UBE3A-Glu+TTX + APV: $n = 12$, GFP + UBE3A-Glu+TTX + APV + CNQX: $n = 13$, GFP + UBE3A + SARNP-TTX + APV: $n = 12$, GFP + UBE3A + SARNP-Glu+TTX + APV: $n = 14$, GFP + UBE3A + SARNP-Glu+TTX + APV + CNQX: $n = 14$ images of 2.25 mm$^2$ area. (C) Representative traces of AMPAR mEPSCs obtained from whole-cell recordings at a holding potential of −70 mV. (D, E, F) Plots of mean amplitude (D), frequency (E) and Decay Tau (F) of mEPSCs recorded from DIV14-17 primary cortical neurons infected with GFP or UBE3A virus, together with SARNP or GluA1 virus for potential rescue. UBE3A overexpression impaired the amplitude of AMPAR mEPSCs, which was rescued by either SARNP or GluA1 restoration. GFP: $n = 7$, UBE3A: $n = 9$, UBE3A + SARNP: $n = 8$, UBE3A +GluA1: $n = 8$. Data information: Mean ± SEM. *$p < 0.05$; **$p < 0.01$; ***$p < 0.001$; ns = not significant. In (B), two-way ANOVA with Bonferroni's multiple comparisons test. In (D, E, F), one-way ANOVA with Bonferroni's multiple comparisons test.

sufficient SARNP expression in the hypothalamus. These possibilities will be clarified by future studies.

# Methods

## Animal care and use

*Ube3a* conventional BAC overexpressing mouse models (*Ube3a*-Tg 2x) on the FVB background (stock #019730) (Smith et al, 2011; Krishnan et al, 2017) and FVB/NJ WT mice (stock #001800) were purchased from the Jackson Laboratory. All animal procedures were conducted in compliance with the policies of the Institutional Animal Care and Use Committee (IACUC) at Boston University (#PROTO201800574), and performed in compliance with the NIH Guide for the Care and Use of Laboratory Animals. *Ube3a* 2x transgenic (2xTg) animals were obtained by breeding heterozygous (*Ube3a* 1xTg) males and heterozygous females, and both male and female homozygous mice, including WT and *Ube3a* 2xTg mice, were used in the experiments.

## Immunohistochemistry of brain slices

Brain slices were prepared for immunohistochemistry as previously described (Gilbert et al, 2020). Prior to brain removal, mice were perfused with ice-cold PBS. The brains were then fixed in 4% paraformaldehyde and cryoprotected in 10% sucrose/PBS for 4–6 h and then cryoprotected in 30% sucrose/PBS until they sank to the bottom. To prepare for sectioning, the dehydrated brains were frozen in optimal cutting temperature-embedded medium (Tissue-Tek, cat. #25608-930) and sliced into 20 µm sections using a CM1850 cyrostat (LEICA Biosystems) at −20 °C. Sections were immediately mounted onto SuperFrost microscope slides (Fisher Scientific, cat.# 12-550-15) and stored at −80 °C for future use. To prepare for immunostaining, sections were rinsed with PBS for at least 2 h. Sections were then blocked in 5% goat serum, 0.3% Triton X-100/PBS for 1 h at room temperature. Properly diluted primary antibodies were applied overnight at 4 °C to target specific proteins of interest. The following day, sections were washed three times with PBS and incubated with the appropriate Alexa Fluor-conjugated secondary antibodies (1:250, Life Technologies) for 1 h at room temperature. Brain slices were then washed three times with PBS, with the first wash containing Hoescht (1:2000, Thermo Fisher Scientific, cat. # 62249). Sections were then mounted with ProLong Gold anti-fade mounting medium (Invitrogen, cat. #P36930), left to dry at room temperature overnight, and stored at −20 °C in dark before imaging.

## Primary neuron culture

Primary cortical neurons were prepared from embryonic day 18 rat fetuses or postnatal day 0 mouse pups (Gilbert et al, 2020). Timed pregnant Sprague–Dawley rats were obtained from Charles River Laboratories Inc, and WT and *Ube3a* 2xTg mouse pups were generated by breeding heterozygous (*Ube3a* 1xTg) males and females. Cortical brain tissues were dissected and digested with papain (0.5 mg/mL in Hanks balanced salt solution, Sigma-Aldrich; cat. #4762) for 20 min at 37 °C. The digested tissues were then gently triturated in a trituration buffer containing 0.1% DNase (cat. #PA5-22017 RRID: AB_11153259), 1% ovomucoid (Sigma-Aldrich; cat. #T2011), 1% bovine serum albumin (Sigma-Aldrich; cat. #05470) in DMEM to fully dissociate neurons. Dissociated neurons were counted and plated onto 18 mm circular coverslips (Carolina Biology Supply, cat. #633013). Coverslips had been coated in poly-l-lysine (Sigma-Aldrich; cat. #P2636; 100 µg/ml in borate buffer) overnight at 37 °C, then washed three times with sterile deionized water and left in plating medium [DMEM supplemented with 10% fetal bovine serum (Atlanta Biologicals; cat. #S11550), 5% horse serum (Atlanta Biologicals; cat. #S12150), 31 mg of l-cysteine, 1% penicillin/streptomycin (Corning; cat. #30–002-Cl), and l-glutamine (Corning; cat. #25–005-Cl)] before cell plating. The day after plating, plating medium was replaced by feeding medium [neurobasal medium (Gibco; cat. #21103049) supplemented with 2% Neurocult SM1 Neuronal Supplement (StemCell Technologies; cat. #05711), 1% horse serum (Atlanta Biologicals; cat. #S12150), 1% penicillin/streptomycin (Corning; cat. #30-002-CI), and L-glutamine (Corning; cat. # 25-005-CI)]. One week after plating, 5-fluorodeoxyuridine (10 µM; Sigma-Aldrich; cat. #F0503) was added to the medium to inhibit glial growth. All cells were maintained in a humidified incubator containing 5% $CO_2$ at 37 °C.

## Immunocytochemistry

Culture neurons were washed twice in ice-cold PBS and fixed in a 4% paraformaldehyde solution at room temperature for 10 min. Cell membranes were then permeabilized for 8 min in 0.3% Triton X-100 (Fisher Biotec; cat. #BP151-100) in PBS. Cells were rinsed three times with PBS, and blocked with 10% goat serum for 1 h. Primary antibodies (in 5% goat serum PBS) were applied overnight at 4 °C, followed by washing twice with PBS and incubated with the appropriate Alexa Fluor-conjugated fluorescent secondary antibodies (1:400, Life Technologies) for an additional hour. Cells were then rinsed (with Hoechst 1:2000 in the first rinse) and mounted on microscopy glass slides using Prolong Gold anti-fade mounting

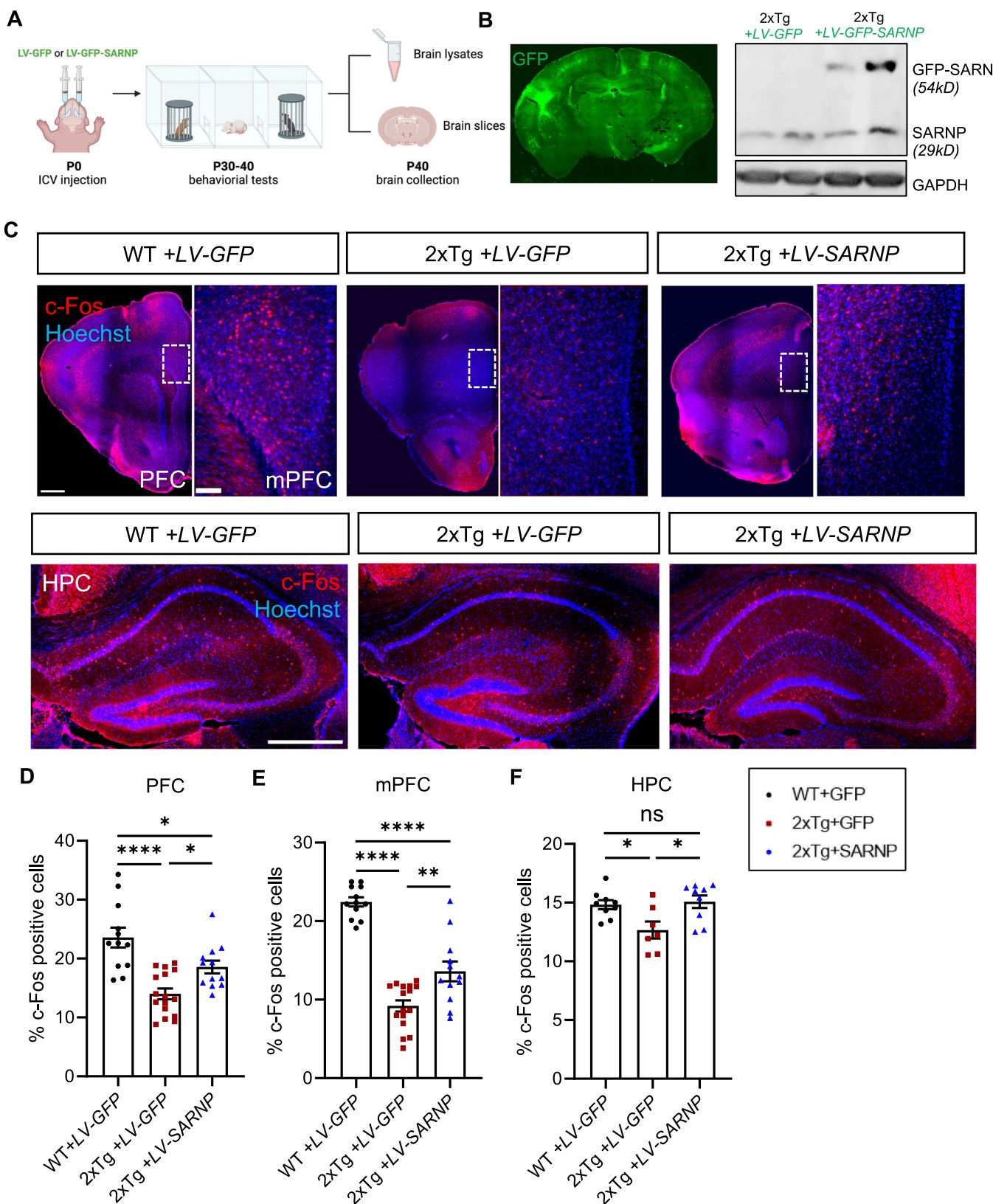

**Figure 6.   Intraventricular brain injection of LV-GFP-SARNP at P0 rescues neuronal activity in *Ube3a* 2xTg mice.**

(A) Schematic illustration of the procedures. LV-GFP or LV-GFP-SARNP virus was injected into the lateral ventricles of mouse brains at P0. Behavioral tests were performed between P30 to P40. Mouse brains were perfused around P40 for cryostat sectioning. (B) Left: viral expression was detected in the whole brain around P40. Right: expression of LV-GFP-SARNP in P40 *Ube3a* 2xTg brains. (C) Immunohistochemistry of c-Fos in prefrontal cortex (PFC, top panel, left side of each condition), medial prefrontal cortex (mPFC, top panel, right side of each condition) and hippocampus (HPC, bottom panel) 40 days after ICV injection of LV-GFP-SARNP. Scale bars, 500 μm (PFC); 500 μm (HPC); 100 μm (mPFC). (D, E) Quantification of the percentage of c-Fos positive cells showed partial rescue of neuronal activity in PFC (D) and mPFC (E) by LV-GFP-SARNP 40 days after ICV injection. WT + LV-GFP: $n = 12$ slices from 3 brains, *Ube3a* 2xTg+LV-GFP: $n = 16$ slices from 4 brains, *Ube3a* 2xTg+LV-SARNP: $n = 12$ slices from 3 brains. (F) Quantification of the percentage of c-Fos positive cells showed full rescue of neuronal activity in HPC by LV-GFP-SARNP 40 days after ICV injection. WT + LV-GFP: $n = 9$ slices from 3 brains, *Ube3a* 2xTg+LV-GFP: $n = 7$ slices from 3 brains, *Ube3a* 2xTg+LV-SARNP: $n = 9$ slices from 3 brains. Data information: Mean ± SEM. $*p < 0.05$; $**p < 0.01$; $****p < 0.0001$; ns = not significant. One-way ANOVA with Bonferroni's multiple comparisons test.

reagent (Invitrogen; cat. #P36930) and stored in the dark at 4 °C for subsequent imaging.

## Neuronal transfection

To transfect neurons with target plasmids, Lipofectamine 2000 (Invitrogen, cat. #11668019) was used according to the manufacturer's instructions on DIV10. Each coverslip was transfected with 1 μg of GFP, or HA-UBE3A (Addgene, cat. #8648), or HA-UBE3A and SARNP (Addgene, cat. #34822), together with 1 μl Lipofectamine 2000. The plasmid and Lipofectamine 2000 were diluted separately in 50 μl DMEM in 1.5 ml Eppendorf tubes and incubated for 5 min. Then solutions from two tubes were then mixed and incubated at room temperature for another 20 min to form the transfection complex. Thereafter, the transfection complex was gently added dropwise to the 12-well plate with 1 coverslip/well. The neurons were incubated with the transfection complex at 37 °C for 3 h before the medium was replaced with feeding medium. On DIV14 (4 days following transfections), neurons were fixed for immunocytochemistry or lysed for biochemical analysis.

## Western blot

The cultured neurons and mouse brain tissues were prepared for Western blot analysis. The cultured neurons were lysed in 2× Laemmli sample buffer (4% SDS, 10% 2-mercaptoethanol, 20% glycerol, 0.004% bromophenol blue, 0.125 M Tris HCl) and boiled at 95 °C for 10 min before being stored at −20 °C for later use. The mouse brain tissues were dissected on ice immediately after sacrificing animals, and then lysed on ice using radioimmunoprecipitation assay (RIPA) lysis buffer (50 mM Tris-HCl, pH 7.4, 150 mM NaCl, 1% NP-40, 1% sodium deoxycholate, 1% sodium dodecyl sulfate) supplemented with cOmplete protease inhibitor (Roche; cat. #11873580001). The lysates were further solubilized by sonication and 1 h rotation at 4 °C. After centrifugation at $12,000 \times g$ for 20 min at 4 °C, the supernatants were collected and subjected to a BCA assay according to the manufacturer's protocol (Pierce, cat. #23225) to determine protein concentrations. Protein levels were normalized using RIPA lysis buffer. The samples were then mixed with 2× Laemmli sample buffer and boiled at 95 °C for 10 min to prepare for SDS-PAGE.

Standard SDS-PAGE procedures were used to separate proteins of interest in the cell or brain lysates. The proteins were then transferred to PVDF membranes (Bio-Rad, cat. #1620177) and blocked in 5% milk in PBS for 1 h. The membranes were then probed for different targets with the appropriate primary antibodies overnight at 4 °C in Tris-buffered saline supplemented with 0.05%

Tween (TBST). After being washed three times in TBST, the membranes were incubated with the appropriate secondary antibody for 1 h. Following another three washes with TBST, the immunoblots were visualized using a chemiluminescence detection system (Sapphire Biomolecular Imager, Azure biosystems, CA, USA) and analyzed using ImageJ.

## RNA fluorescent in situ hybridization

DNA probes were generated by nick translation using Nick Translation Kit (Roche, cat. #10976776001) and Amersham CyDye Fluorescent Nucleotides (Cytiva, cat. #PA53022), following the manufacturer's instructions.

Culture neurons were washed twice in ice-cold PBS and fixed in a 4% paraformaldehyde solution at room temperature for 10 min. Cell membranes were then permeabilized for 8 min in 0.3% Triton X-100 (Fisher Biotec; cat. #BP151-100) in PBS. Cells were rinsed three times in PBS, followed by one 10 min rinse in 2× SSC (Sigma-Aldrich, cat. # S6639). Nick-translated probes were denatured by heating at 90 °C for 10 min. A hybridization solution consisting of 50% formamide (Sigma-Aldrich, cat. #F9037), 2x SSC, 10% dextran sulfate (Sigma-Aldrich, cat. #D8906), 10 μg/ml yeast tRNA (Invitrogen, cat. #AM7119) and nick-translated probes was prepared, and 20 μl of the solution was placed onto a slide as a single drop. Coverslips with neurons were removed from the 2x SSC buffer and inverted onto the drop of hybridization solution with the cells facing the hybridization mix. Neurons were then incubated in a dark humid chamber at 37 °C for 12–16 h. Neurons were washed three times at 42 °C in freshly made 50% formamide/2x SSC, followed by three rinses in 2× SSC at 42 °C, with Hoechst 1:2000 in the last wash to label the nucleus. Finally, the coverslip was mounted to microscopy glass slides with Prolong Gold antifade mounting reagent (Invitrogen; cat. #P36930) and stored at 4 °C in dark for subsequent imaging. Nuclear GluA1 mRNA was assessed by its colocalization with the Hoechst nuclear dye. The remaining signal in the soma was measured to quantify the levels of cytosolic GluA1 mRNA.

## Real-time PCR

The prefrontal cortical area was used to extract total RNA using the RNeasy® Mini Kit (Qiagen, cat. # 74104) following the manufacturer's instructions. cDNA was reverse transcribed from 500 ng of total RNA for each sample using HiFiScript cDNA Synthesis Kit (CoWin Biosciences, cat. # CW2569M) according to the manufacturer's protocol. cDNA samples were then used as a template for further mRNA level analysis. For qPCR, 1 μl of the reverse

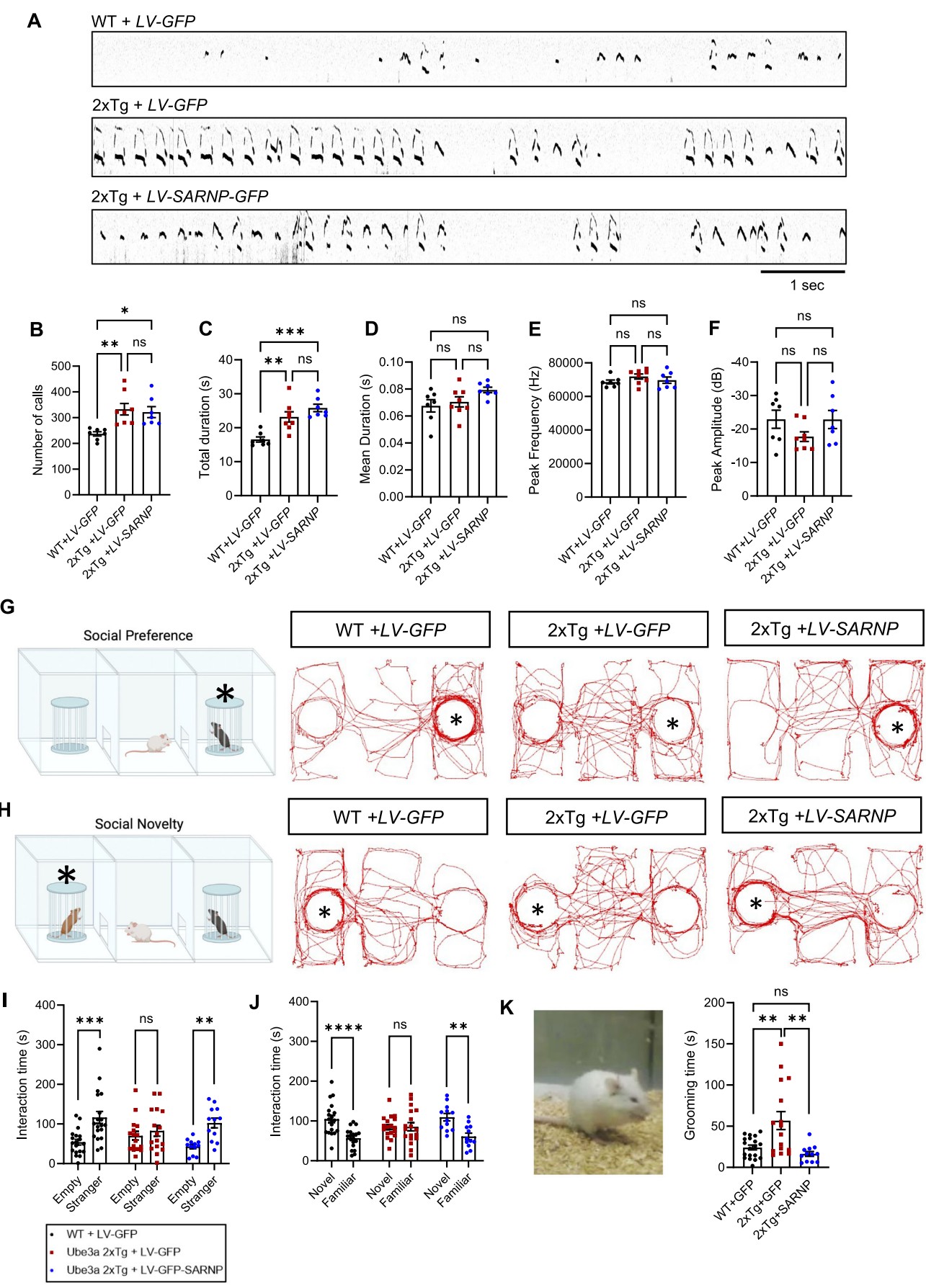

**Figure 7. Intraventricular brain injection of LV-GFP-SARNP at P0 rescues social interaction and self-grooming behaviors in *Ube3a* 2xTg mice.**

(A) Representative vocalization recordings from P9 mice. (B–F) Quantification of the number of calls (B), total call duration (C), mean call syllable duration (D), peak frequency (E), and peak amplitude (F). (G, H) The paradigm for the three-chamber social test and traces of track paths. For social preference (G), an unfamiliar mouse was placed into either of the side chambers and the test mouse was allowed to move freely in the apparatus. For social novelty (H), a second mouse (novel mouse) was placed into the remaining empty chamber, and the test mouse was allowed to interact with both mice. (I) Quantification of the interaction time with empty cage vs. stranger mouse showed a decrease in preference for the stranger mouse in *Ube3a* 2xTg mice, which was rescued by ICV injection of LV-GFP-SARNP. (J) *Ube3a* 2xTg mice displayed abnormal preference for the novel mouse, which was rescued by ICV injection of LV-GFP-SARNP. (K) *Ube3a* 2xTg mice showed an increase in grooming time, which was rescued by ICV injection of LV-GFP-SARNP. Data information: In (B–F), $n = 7$ WT + LV-GFP, $n = 8$ *Ube3a* 2xTg+LV-GFP, $n = 7$ *Ube3a* 2xTg+LV-GFP-SARNP. In (I–K), $n = 19$ WT + LV-GFP, $n = 16$ *Ube3a* 2xTg+LV-GFP, $n = 12$ *Ube3a* 2xTg+LV-GFP-SARNP. Mean ± SEM. $*p < 0.05$; $**p < 0.01$; $***p < 0.001$; $****p < 0.0001$; ns = not significant. In (B–F, K), One-way ANOVA with Bonferroni's multiple comparisons test. In (I, J), two-way ANOVA with Bonferroni's multiple comparisons test.

transcribed cDNA was used as a template and mixed with primer pairs:

GluA1 (5′-ACCCTCCATGTGATCGAAATG-3′, 5′-GGTTCTATTCTGGACGCTTGAG-3′);

GAPDH (5′-TGAGGCCGGTGCTGAGTATGTCG-3′, 5′-CCACAGTCTTCTGGGTGGCAGTG-3′);

U6 (5′-CTCGCTTCGGCAGCACA-3′, 5′-AACGCTTCACGAATTTGCGT-3′);

Ten μl reaction mixture was prepared with UltraSYBR Mixture kit (CoWin Biosciences, cat. #CW2602F) according to the manufacturer's protocol, and then put in the Applied Biosystems 7900HT Fast Real-Time PCR system for real-time monitoring.

## shRNA construction

Two shRNA constructs were created to target two sequences of SARNP mRNA:

#1: 5'-TCTGCCTACATACACAGTCAT-3'
#2: 5'-GCACAGAGATTTGGTTTGAAT-3'

The shRNA sequences were inserted into the pLKO.3G cloning vector (Addgene; cat. # 14748) using PacI and EcoRI restriction sites, following the manufacturer's instructions. To knock down SARNP in neurons, shRNA #1 and shRNA #2 were mixed and transfected into the neurons.

## Immunoprecipitation

For puromycin assay, primary cortical neurons were infected with either GFP virus, UBE3A virus, or both UBE3A and SARNP virus at DIV4. At DIV11, neurons were treated with either DMSO or puromycin for 12 h.

For ubiquitination immunoprecipitation assays, HEK cells were transfected with GluA1, HA-Ubiquitin, HA-UBE3A or ligase dead mutant UBE3A-C820A using the polyethylenimine (PEI) transfection reagent (Polysciences; cat. #23966) with a 3:1 PEI-to-DNA ratio when the cells reached 70% confluency. HEK cells were lysed and collected 48 h post-transfection.

Mg132 (10 μM) was added to the cultured neurons 6 h before collecting lysates. Cultured cortical neurons or HEK cells were rinsed with cold PBS and lysed on ice in 200 μl of RIPA lysis buffer (50 mM Tris-HCl, pH 7.4, 150 mM NaCl, 1% NP-40, 1% sodium deoxycholate, 1% sodium dodecyl sulfate) supplemented with cOmplete protease inhibitor (Roche; cat. #11873580001). Cell lysates were further solubilized by sonication and 1 h rotation at 4 °C. After rotation, lysates were centrifuged at $12,000 \times g$, 20 min, 4 °C to collect supernatant. A small portion of each sample was saved as total cell lysate while the rest was adjusted to 500 μl with

more RIPA buffer for immunoprecipitation. Protein A-Agarose beads (Santa Cruz Biotechnology; cat. #sc-2001) were added to the lysates along with antibodies against GluA1 N-terminal (Alomone labs, cat. #AGC-004). Samples were incubated overnight for 12–16 h on rotation at 4 °C. Agarose beads were then washed three times with ice-cold RIPA buffer, resuspended in 2× Laemmli sample buffer, and denatured at 95 °C for 10 min before being subjected to western blotting.

## Electrophysiology

AMPAR-mediated miniature excitatory postsynaptic currents (mEPSCs) were recorded from primary cortical neurons infected with virus expressing GFP or UBE3A together with GFP, SARNP or GluA1. Cells were recorded at $-70$ mV using a cesium-based intracellular solution (100 mM Cs-methanesulfonate, 10 mM CsCl, 10 mM HEPES, 0.2 mM EGTA, 4 mM Mg-ATP, 0.3 mM Na-GTP and 10 mM Na-phosphocreatine, pH 7.4). The extracellular solution (140 mM NaCl, 3 mM KCl, 1.5 mM MgCl₂, 2.5 mM CaCl₂, 11 mM Glucose, 10 mM HEPES, pH 7.4) was supplemented with 50 μM APV, 20 μM bicuculline, and 1 μM TTX. Continuous data were recorded using an Axopatch 200B amplifier, an Axon Digidata 1440 A acquisition board, and pClamp software (version 10.7, Molecular Devices). Signals were filtered at 1 Hz and sampled at 20 kHz. 5 min of synaptic events were recorded, and the amplitude and frequency of AMPAR-mediated currents were analyzed offline with pClamp software (version 10.7, Molecular Devices), using amplitude as the primary identification parameter and a 5 pA cut-off to account for noise. The amplitude and the interevent interval were collected.

## Viral constructs preparation and virus infection

To produce AAV-UBE3A, UBE3A/E6AP was cut from the p4054 E6AP plasmid (Addgene, cat. #8658) and cloned into AAV-ReaChR-citrine (Addgene, cat. #50954), as described in our previous work (Khatri et al, 2018). The resulting construct is in an AAV2 vector with UBE3A expression driven by a human synapsin promoter. For AAV particle production, HEK293T cells were transfected with the AAV-UBE3A or AAV-GFP plasmid, along with viral packaging and envelope proteins XX6.80 and XR2, using polyethylenimine (Polysciences; cat. #23966). Three days post-transfection, cells were lysed by freeze–thaw cycles and sonication. The lysate was centrifuged at $3000 \times g$ for 30 min at 4 °C. The supernatant was filtered through a 0.45 μm filter and precipitated with PEG-it (1:5, System Biosciences; cat. #LV810A-1). The mixture was then centrifuged at 1500 rpm for 30 min, and the

resulting viral pellet was resuspended in PBS and aliquots were stored at −80 °C.

The pQTEV-CIP29/SARNP plasmid (Addgene, cat. #34822) was used to extract SARNP, which was then inserted into the pEGFP-C1 vector to create the pEGFP-SARNP construct. GFP-SARNP was then subcloned into pFUW lentiviral (LV) vector (Addgene, cat. #14882), to generate pFUW-GFP-SARNP construct, with GFP-SARNP expression driven by a CMV promoter. To package the VSV-G pseudotyped LV particles, HEK293T cells at 70% confluency in 15-cm culture plates were transfected with pFUW-GFP-SARNP, along with viral packaging and envelope proteins (pRSV/REV, pMDLg/RRE, and pVSVG) using polyethylenimine transfection reagent (Polysciences; cat. #23966). 48 h after transfection, cell medium was filtered through a 0.45 µm filter, and PEG-it virus precipitation solution (1:5, System Biosciences; cat. #LV810A-1) was added and kept at 4 °C overnight. On the next day, the solution was centrifuged (1500 rpm, 30 min, 4 °C), and the pellet was resuspended with 100 µl cold PBS. The virus was aliquoted and stored at −80 °C until use.

Viral particle titers were determined using qPCR with SYBR green. The titers were calculated based on the Ct values and a standard curve. For in vitro assays, neurons were infected with AAV at DIV3 and with lentivirus at DIV7, and the medium was replaced with feeding medium 3 days after infection.

## Intracerebroventricular brain injection of virus in neonatal mice

On postnatal day 0, bilateral intracerebroventricular (ICV) injections were carried out in neonatal mouse pups. The pups were first cryoanesthetized and then placed on a cold metal plate. To perform the injection, a 30-gauge needle (Hamilton, cat. #7803-04) and 10 µl Hamilton syringe (Hamilton, cat. #7653-01) were utilized. A separate needle was used to puncture the skull at a site approximately two-fifths of the distance between the lamda and the eye. Each cerebral ventricle was slowly injected with either 2 µl of pFUW-GFP or pFUW-GFP-SARNP virus. After injection, the neonatal mice were returned to their parents until they were weaned. Beginning on postnatal day 30, all the mice underwent behavioral assessments before being euthanized for biochemical and histological analysis.

## Behavioral tests

### Three-chamber social test
Three-chamber social test was performed as described in our previous study (Gilbert et al, 2020). A three-chambered box was made from plastic board that is 0.75 inches thick and measures 65 × 28 × 28 cm. The walls to the center chamber had cut-out doors that are 10 × 10 cm in size allowing movement between chambers. Small wire cages were placed in the two side chambers to house social stranger and/or novel mice. Before the test, mice were habituated to the apparatus with empty cages in both side chambers for 3 days, with each session lasting 5 min, allowing them to move freely within the apparatus. For the Social Preference Test, mice were singly placed into the middle chamber, with the doors blocked by plastic boards and a stranger mouse was placed into one of the side chambers under the wire cage. The doors were unblocked, and the test mouse was allowed to move freely within the apparatus for

5 min. In the Social Novelty Test, the test mouse was returned to the middle chamber, the doors were blocked again. A second mouse (novel mouse) was introduced into the other side chamber. The center doors were unblocked and the test mouse was allowed to move freely within the apparatus for another 5 min. To eliminate odor cues between animals, the entire apparatus was wiped with 70% ethanol between mice. Video recordings were captured with a Logitech c920 webcam during the test and the time spent interacting with each mouse or empty cage (nose ≤2 cm) and locomotion tracks were scored using TrackMo (https://github.com/zudi-lin/tracking_toolbox), blinded to the genotype.

### Ultrasonic vocalization (USV) recording
Ultrasonic vocalization recording of pup isolation calls was performed as described in our previous work (Gilbert et al, 2020). On the test day, P9 pups were separated from the parents and littermates, and isolated in a sound-proof cylindrical plastic container at room temperature (~22 °C). Mouse vocalizations were recorded on postnatal day 5 in a random order for each litter, using a CM16/CMPA microphone (Avisoft Bioacoustics) positioned 15 cm above the pups. The recording chamber was cleaned with 70% ethanol and dried between each pup. USV signals were recorded at a sampling rate of 300 kHz in 16-bit format. The microphone was connected to a preamplifier UltraSoundGate 116Hb (Avisoft Bioacoustics) and digitized sonograms were stored in a computer. SASLab Pro 5.2.12 (Avisoft Bioacoustics) was used to analyze the recordings. Spectrograms were generated using fast Fourier transform (256 FFT length, 100% frame, FlatTop window, and 50% window overlap) and a high-pass filter was applied to eliminate background noise <25 kHz. USVs were automatically detected (threshold: −47 ± 10 dB, hold time: 7 ms) followed by manual inspection to ensure accuracy. Several acoustic parameters of calls were measured including number of calls, mean call duration, total time spent calling, peak frequency, and peak amplitude.

### Self-grooming behavior test
The self-grooming behavior test was performed as described in our previous work (Huo et al, 2022). Each test mouse was placed individually into a regular housing cage with a fresh thin layer (~0.5 cm) of wood chip beddings and was allowed to acclimate for 20 min. Afterwards, the mice were video recorded for 10 min from a side view of the cage using a camera. The total time spent on grooming was measured.

## Statistical tests

The statistical analysis and graphic presentations were performed using GraphPad Prism 8.0 (GraphPad Software). Data of genotypes were analyzed using an unpaired two-tailed Student's $t$ test, with genotype as a factor. For comparisons involving more than two levels of transfection, drug treatment, or virus infection, we used one-way analysis of variance (ANOVA). Two-way ANOVA was used to analyze comparisons involving two independent variables, each with at least two levels, such as different treatment conditions or cytosol versus nucleus between WT and Tg. Post hoc multiple comparisons were corrected using Bonferroni's method. Following statistical analysis, individual data points with a value larger or smaller than 2 standard deviations were considered outliers and

removed from the data pool. All results are presented as mean ± SEM (standard error of mean), and $*p < 0.05$ was considered statistically significant.

## Data availability

Source data for all figures are available at the Harvard Dataverse repository: https://doi.org/10.7910/DVN/WSFXPH.

## Peer review information

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

## Acknowledgements

We would like to thank the Man Lab members for helpful discussion. We thank Ameya Patkar for his comments on the manuscript. This work was supported by NIH grants R01 MH079407, R01 MH130600, R21 MH133014.

## Author contributions

Yuan Tian: Conceptualization; Data curation; Software; Formal analysis; Validation; Investigation; Visualization; Methodology; Writing—original draft; Writing—review and editing. Feiyuan Yu: Software; Investigation; Methodology. Eunice Yun: Investigation; Methodology. Jen-Wei Lin: Validation; Writing—original draft; Writing—review and editing. Heng-Ye Man: Conceptualization; Resources; Supervision; Funding acquisition; Validation; Investigation; Methodology; Writing—original draft; Project administration; Writing—review and editing.

## Disclosure and competing interests statement

The authors declare no competing interests.

# Expanded View Figures

**Figure EV1.  Validation of GluA1 mRNA FISH.**

(**A**) GluA1 RNA FISH (red) in DIV14 primary rat neurons transfected with either SCR-GFP, shGluA1-GFP to knock down GluA1, or GluA1-GFP to overexpress GluA1. Hoechst stained nucleus. Arrows point to the transfected cells. Scale bars, 25 μm. (**B**) Quantification showed a reduction in the FISH fluorescent intensity of GluA1 mRNA following GluA1 knock down by shRNA; and increased FISH fluorescent intensity of GluA1 mRNA following GluA1 overexpressing. (**C**) Knocking down or overexpressing GluA1 mainly affected GluA1 mRNA level in the cytosol rather than that in the nucleus. Data information: SCR-GFP: $n = 35$, shGluA1-GFP: $n = 36$, GluA1-GFP: $n = 33$. Mean ± SEM. **$p < 0.01$; ***$p < 0.001$; ****$p < 0.0001$; ns = not significant. In (**B**), one-way ANOVA with Bonferroni's multiple comparisons test. In (**C**), two-way ANOVA with Bonferroni's multiple comparisons test.

▶

                                 

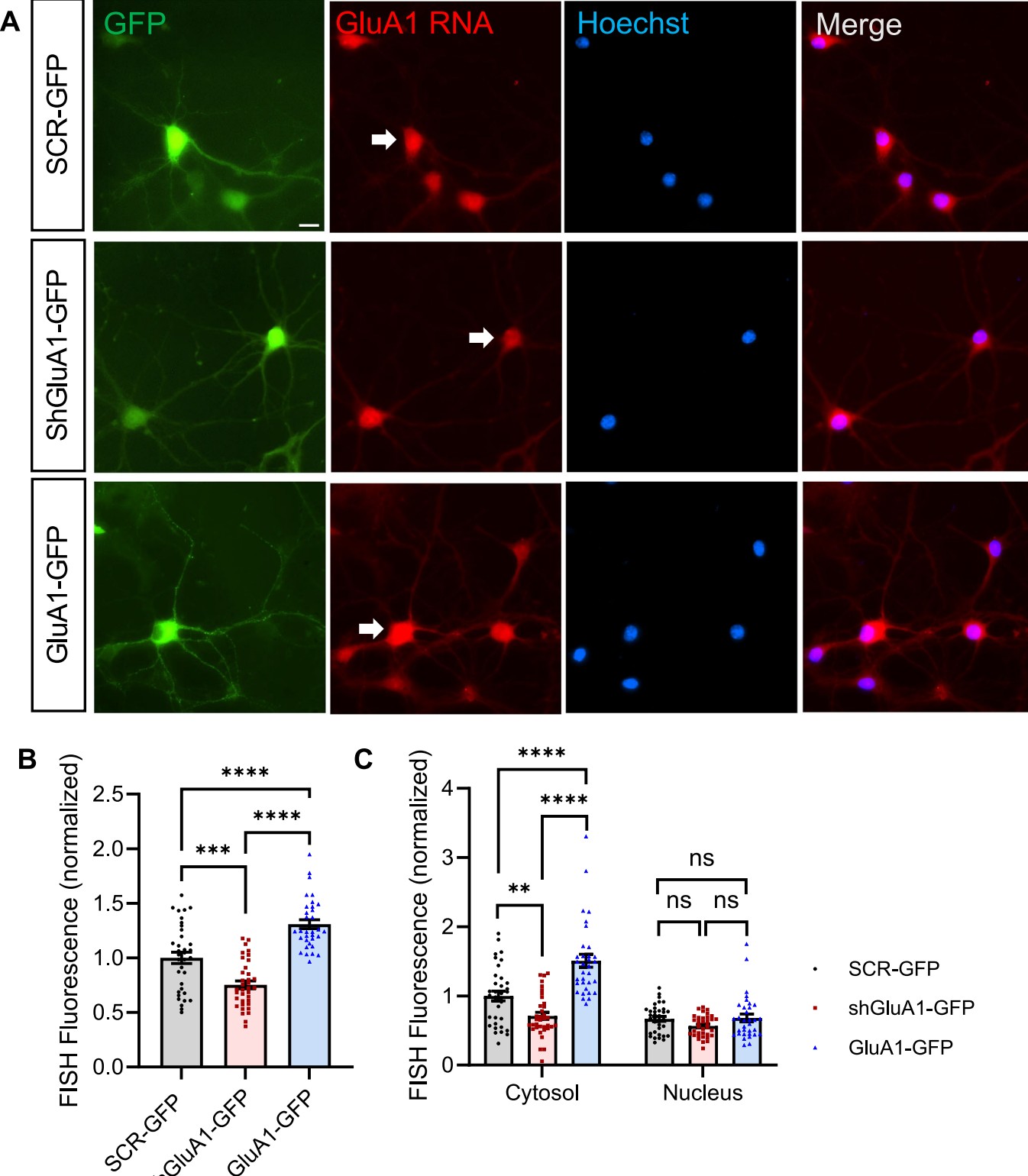

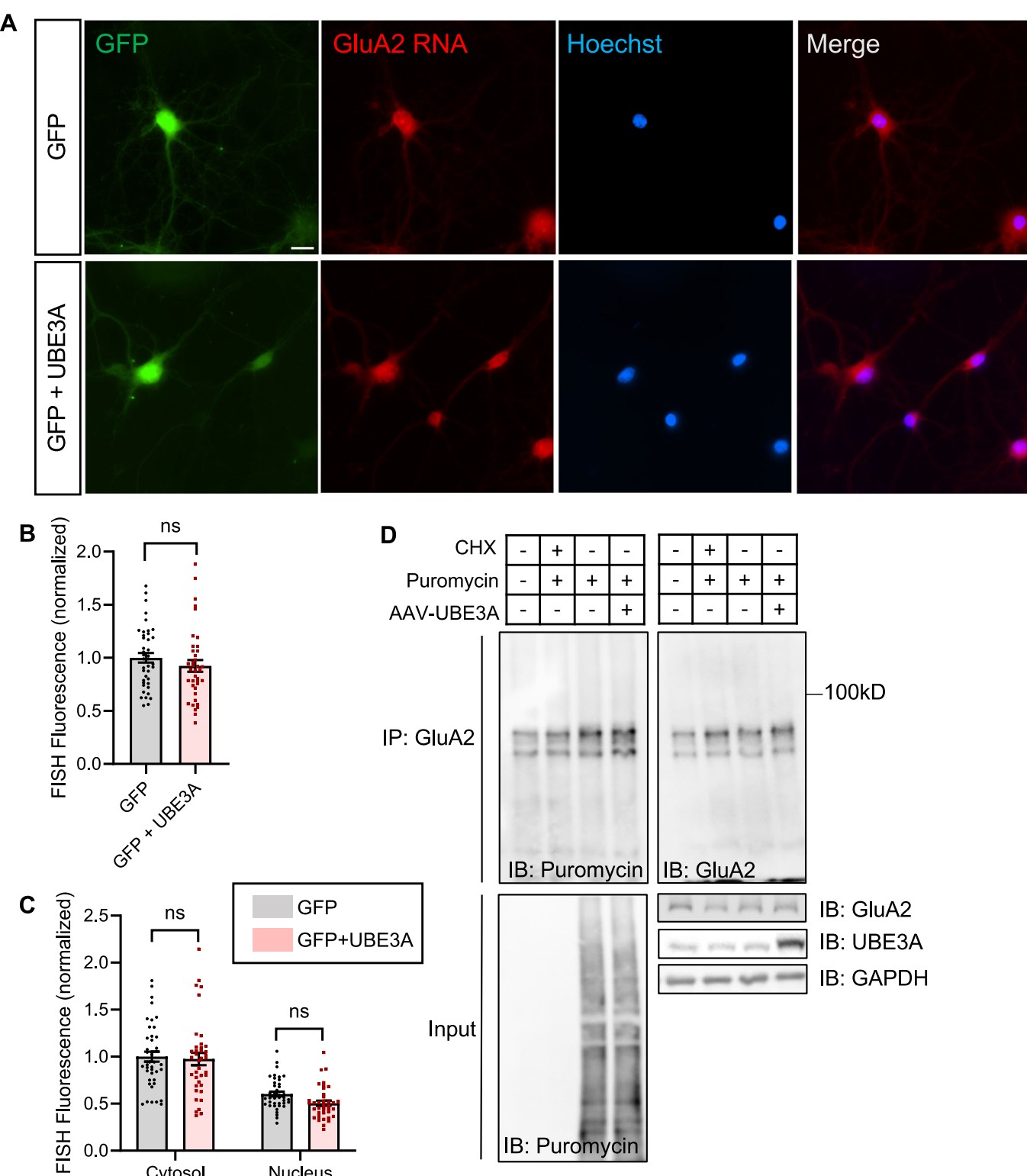

◄

**Figure EV2. GluA2 mRNA localization and GluA2 protein synthesis are intact with UBE3A overexpression.**

(A) Primary rat neurons were transfected with either GFP or GFP + UBE3A, and GluA2 mRNA (red) was examined by RNA FISH at DIV14. The nucleus was stained with Hoechst. Scale bars, 25 µm. (B, C) Quantification of the GluA2 mRNA intensity showed that UBE3A overexpression did not alter the GluA2 mRNA levels in the soma, nor did it alter GluA2 mRNA distribution in the cytosol vs. nucleus. GFP: $n = 41$, GFP + UBE3A: $n = 39$. (D) Puromycin assay was performed to examine GluA2 protein synthesis in primary cortical neurons infected with GFP or UBE3A virus. The neurons were treated with DMSO or puromycin for 12 h followed by GluA2 immunoprecipitation and probing for puromycin. Cell lysates (input) were also probed to detect total protein levels. Data information: Mean ± SEM. ns = not significant. In (B), unpaired two-tailed *t* test. In (C), two-way ANOVA with Bonferroni's multiple comparisons test.

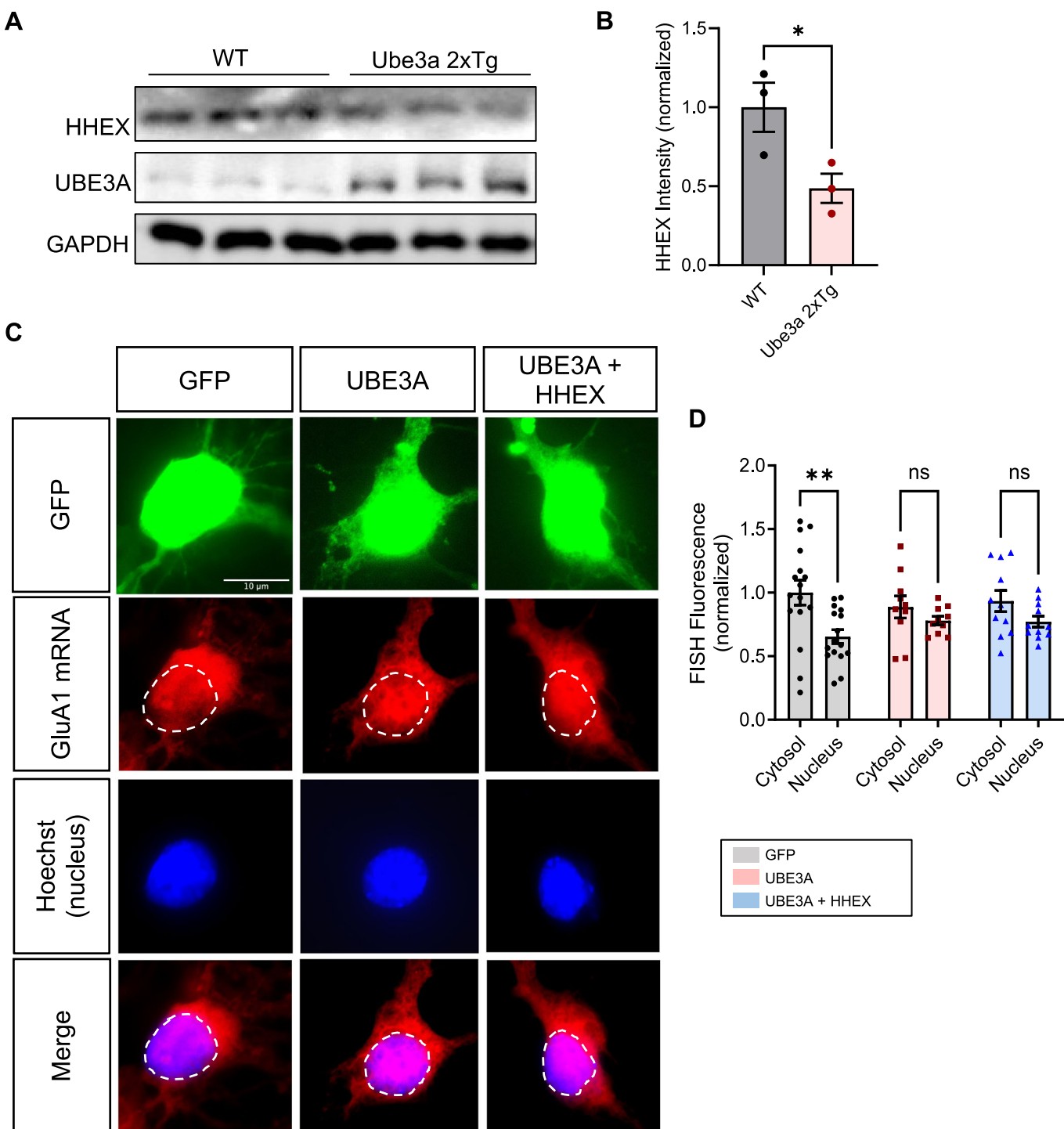

**Figure EV3. Nuclear retention of GluA1 mRNA is not attributed to HHEX reduction in UBE3A-overexpression conditions.**

(A) Western blot analysis revealed HHEX protein expression levels in brain lysates from WT and *Ube3a* 2xTg mice. (B) A notable decrease in HHEX was observed in *Ube3a* 2xTg mouse brains. $n = 3$ animals/group. (C) GluA1 RNA FISH (red) was conducted on rat primary cortical neurons that were transfected with either GFP, GFP + UBE3A, or GFP + UBE3A + HHEX. Scale bars, 10 µm. (D) UBE3A transfection resulted in nuclear retention of GluA1 mRNA, which was not rescued by HHEX co-transfection. GFP: $n = 16$, GFP + UBE3A: $n = 10$, GFP + UBE3A + HHEX: $n = 11$. Data information: Mean ± SEM. $*p < 0.05$; $**p < 0.01$; ns = not significant. In (B), unpaired two-tailed $t$ test. In (D), two-way ANOVA with Bonferroni's multiple comparisons test.

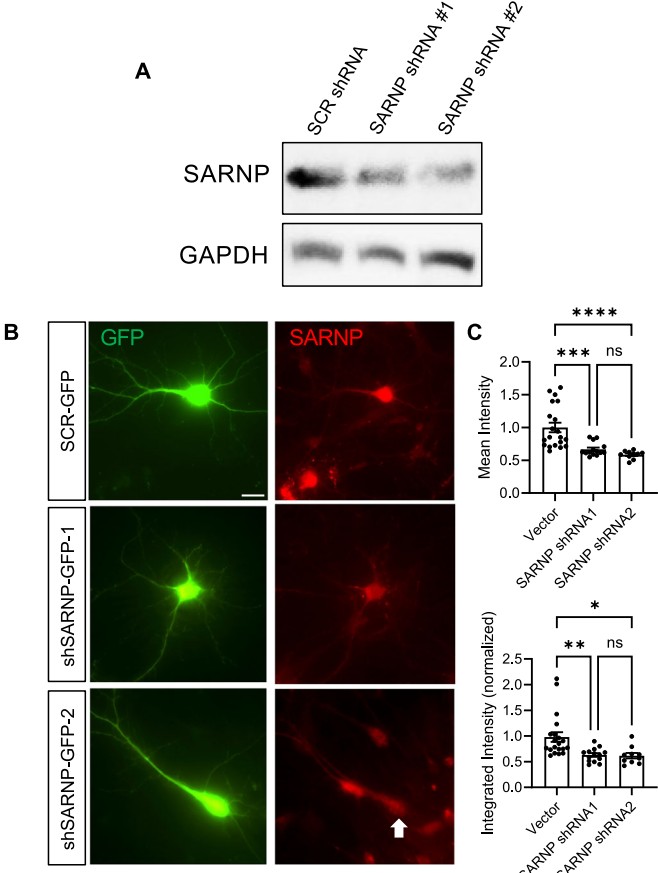

**Figure EV4. SARNP shRNAs effectively reduce SARNP expression in HEK cells and neurons.**

(A) Validation of SARNP shRNAs in HEK cells. HEK cells were transfected with SCR shRNA, SARNP shRNA #1, or SARNP shRNA #2. Both SARNP shRNA constructs reduced SARNP expression. (B) Validation of SARNP shRNAs in primary rat neurons. Neurons were transfected with SCR vector, SARNP shRNA #1, or SARNP shRNA #2. Arrows point to the transfected cells. Scale bars, 25 μm. (C) Mean intensity (top) and integrated intensity (bottom) of SARNP in the soma were significantly reduced by SARNP shRNA #1 and SARNP shRNA #2. SCR vector: $n = 20$, SARNP shRNA #1: $n = 15$, SARNP shRNA #2: $n = 10$. Data information: Mean ± SEM. *$p < 0.05$; **$p < 0.01$; ***$p < 0.001$; ****$p < 0.0001$; ns = not significant. One-way ANOVA with Bonferroni's multiple comparisons test.

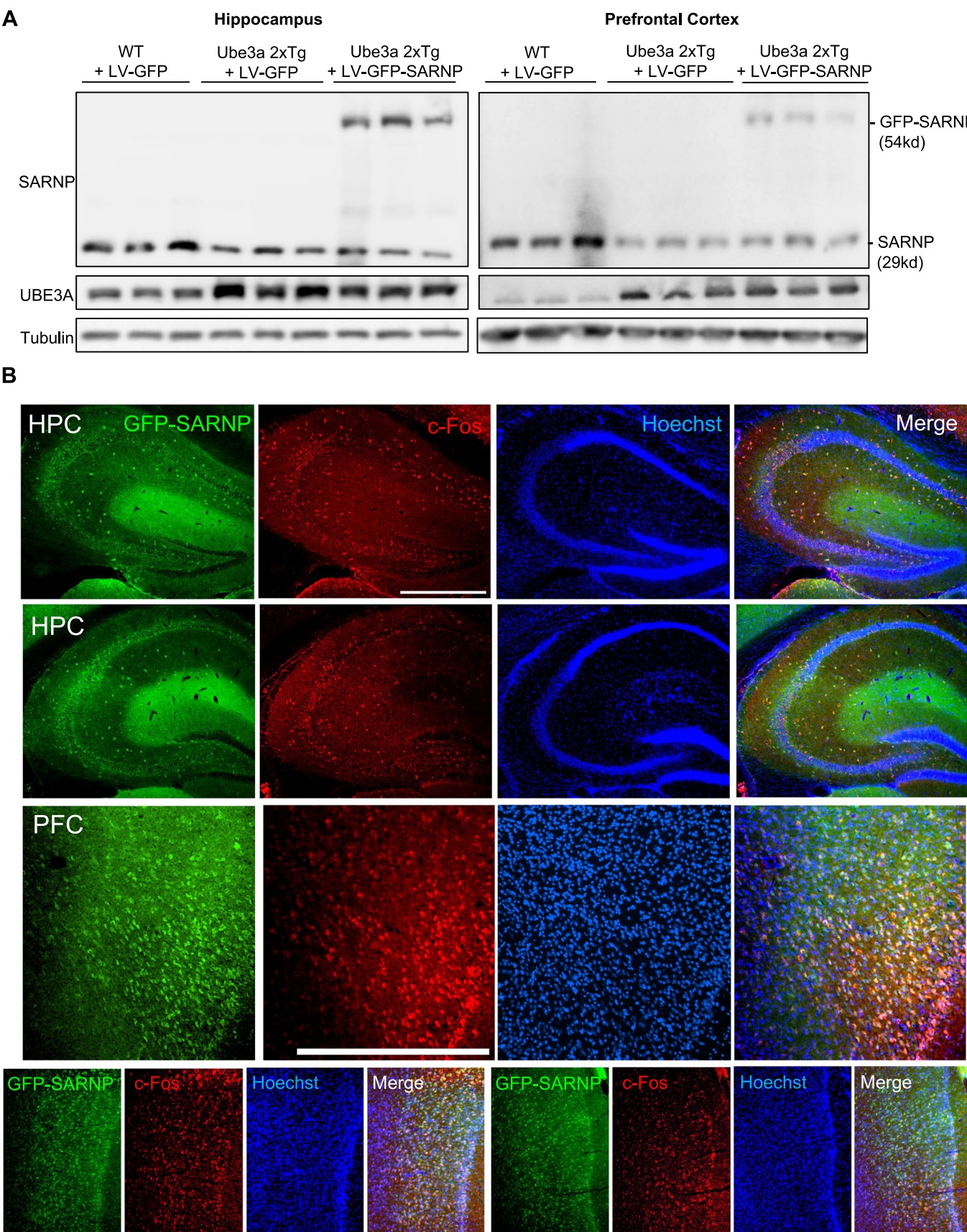

◀ **Figure EV5. LV-GFP-SARNP virus expression in *Ube3a* 2xTg mouse brain.**

(**A**) In P40 *Ube3a* 2xTg mice injected with LV-GFP-SARNP virus, lysates from the hippocampus and prefrontal cortex demonstrated robust GFP-SARNP expression. (**B**) Using immunohistochemistry, colocalization of GFP-SARNP expression with c-Fos was observed in the hippocampus and prefrontal cortex of P40 *Ube3a* 2xTg mice injected with LV-GFP-SARNP. Scale bars, 500 μm.

