## [Peer Review File · EMBO Reports]

mRNA nuclear retention reduces AMPAR expression and promotes autistic behavior in UBE3A-overexpressing mice

Yuan Tian, Feiyuan Yu, Eunice Yun, Jen-Wei Lin, and Hengye Man

Corresponding author(s): Hengye Man (hman@bu.edu)

Review Timeline:

Submission Date:	21st Jun 23
Editorial Decision:	15th Aug 23
Revision Received:	10th Nov 23
Editorial Decision:	4th Dec 23
Revision Received:	7th Jan 24
Accepted:	16th Jan 24

Editor: Esther Schnapp

Transaction Report:

Dear Dr. Man,

Thank you for the submission of your manuscript to EMBO reports. We have now received the full set of referee reports that is pasted below.

As you will see, the referees acknowledge that the findings are potentially interesting. However, they also have several suggestions for how the study could be improved and I think all suggestions are reasonable and should be addressed. Please let me know in case you disagree and we can discuss the revision requirements further, also in a video chat, if you like.

I would thus like to invite you to revise your manuscript with the understanding that the referee concerns must be fully addressed and their suggestions taken on board. Please address all referee concerns in a complete point-by-point response. Acceptance of the manuscript will depend on a positive outcome of a second round of review. It is EMBO reports policy to allow a single round of major revision only and acceptance or rejection of the manuscript will therefore depend on the completeness of your responses included in the next, final version of the manuscript.

We realize that it is difficult to revise to a specific deadline. In the interest of protecting the conceptual advance provided by the work, we recommend a revision within 3 months (15th Nov 2023). Please discuss the revision progress ahead of this time with the editor if you require more time to complete the revisions.

- 1) A data availability section providing access to data deposited in public databases is missing. If you have not deposited any data, please add a sentence to the data availability section that explains that.
- 2) Your manuscript contains statistics and error bars based on $n=2$. Please use scatter blots in these cases. No statistics should be calculated if $n=2$.

5) a complete author checklist, which you can download from our author guidelines <https://www.embopress.org/page/journal/14693178/authorguide>. Please insert information in the checklist that is also reflected in the manuscript. The completed author checklist will also be part of the RPF.

6) Please note that all corresponding authors are required to supply an ORCID ID for their name upon submission of a revised manuscript (<https://orcid.org/>). Please find instructions on how to link your ORCID ID to your account in our manuscript tracking system in our Author guidelines <https://www.embopress.org/page/journal/14693178/authorguide#authorshipguidelines>

7) Before submitting your revision, primary datasets produced in this study need to be deposited in an appropriate public

database (see <https://www.embopress.org/page/journal/14693178/authorguide#datadeposition>). Please remember to provide a reviewer password if the datasets are not yet public. The accession numbers and database should be listed in a formal "Data Availability" section placed after Materials & Method (see also <https://www.embopress.org/page/journal/14693178/authorguide#datadeposition>). Please note that the Data Availability Section is restricted to new primary data that are part of this study. * Note - All links should resolve to a page where the data can be accessed. *

I look forward to seeing a revised form of your manuscript when it is ready.

Yours sincerely,

Referee #1:

The goal of the current research is to determine the neurobiological mechanisms underlying Ube3A-dependent autism spectrum disorders (ASD). Ube3A is known to be an E3 ubiquitin ligase and transcription coactivator. Also, the Ube3A gene is a known ASD risk gene, and increased Ube3A activity is implicated in the development of ASD in humans. The current work examines the neurobiological underpinnings of ASD in the context of Ube3A overexpression, with a particular focus on AMPA receptor (AMPA) malfunction as the cause. The study reveals that the expression of AMPARs and their synaptic distribution are significantly reduced in neurons with UBE3A overexpression and in the brains of Ube3A transgenic mice. Moreover, the team finds that the mRNA of the GluA1 subunit is restricted in the nucleus, and this mRNA retention causes a reduction in GluA1 protein synthesis. Additionally, the current research identifies that SARNP, an mRNA nuclear export protein, is downregulated in neurons overexpressing Ube3A, which is responsible for GluA1 mRNA nuclear retention. Finally, the team also confirms their findings in animal behaviors. Overall, these findings indicate that SARNP plays a crucial role in the cellular and behavioral phenotypes of Ube3A-associated ASD by regulating the nuclear mRNA trafficking and protein translation of the AMPAR GluA1 subunit. The current research topic is significant, and the integrated experimental designs are excellent. However, there are several major issues that are necessary to be addressed.

Major issues

1. Ube3a functions as a transcription coactivator. However, it is unclear how higher activity of Ube3A downregulates SARNP? The authors should address the potential mechanisms.
2. The authors used in vitro and in vivo transfection and infection of Ube3A and SARNP throughout the study. However, there is insufficient data to confirm whether these proteins are expressed. For instance, the authors expressed Ube3A and SARNP using AAV and lentivirus, respectively, in Figs. 4A and 5. Since it is implausible that two separate viruses would be transduced concurrently, it is uncertain if these two proteins were indeed expressed in the same cells. The data demonstrating these expressions should be included.
3. In Fig 6, it is unclear whether SARNP is indeed expressed in prefrontal and hippocampal neurons to reverse the cFos phenotype. The data confirming SARNP expression in these areas should be included.
4. In three chamber experiments, the authors should compare the interaction times within the experimental groups - e.g. WT vs. 2xTg vs 2xTg+SARNP - in addition to the interactions between empty and stranger and between novel and familiar.
5. It is unclear if there is sex difference.
6. The manuscript stated "The cellular and molecular mechanisms underlying the synaptic dysfunction in UBE3A-dependent ASD remain unclear". However, in the previous study (Smith et al., 2011), overexpression of Ube3A in mice shows that synaptic transmission is suppressed because of reduced presynaptic release probability, synaptic glutamate concentration, and postsynaptic action potential coupling, which contributes to ASD. Therefore, how the presented mechanism integrates or acts independently from these described mechanisms should be discussed.

Minor issues

1. The previous study shows that ultrasonic vocalization is regulated by the hypothalamus (Zimmer et al., 2019). Therefore, this discussion can be included.
2. Mice in the FVB background are blind. Therefore, the objective recognition test and other behavioral assessments that rely on visual cues are inappropriate for use with these animals.
3. Insufficient information about viruses (the sources, titers, and etc).

Referee #2:

The manuscript by Tian et al. explores an interesting mechanism linked to overexpression of the autism risk gene UBE3A. While the field focuses on loss of function (Angelman Syndrome), the paper summarizes in the introduction the effects of UBE3A duplication and overexpression studies. The mechanism proposed is clear and the experiments support the claims of the authors. I only have some minor points that would need to be addressed prior to publication:

- 1) The colocalization of GLuA1 mRNA and the nuclear dye (e.g. Fig. 3H, J). This needs to be better described in the

methodology and for some of the images it is not intuitive (apart from the merged image) what the significance of the dotted white circles is.

2) USV- I could not find mention of the temperature of embryos in methods during USV recording.

3) Degradation of GluA1 by UBE3A. While the authors examine mRNA localization and protein synthesis, it would be good to discuss the potential contribution of GluA1 degradation given that the enzyme is a ubiquitin E3 ligase.

Referee #3:

Tian et al reported that the expression of AMPARs and their synaptic distribution were significantly decreased in UBE3A-overexpressing neurons and in Ube3a 2xTg brains. GluA1 mRNA was restricted in the nucleus, which is regulated by SARNP, a mRNA nuclear export factor, in the overexpressing neurons. The authors also presented that restoration of SARNP levels rescued GluA1 expression, neuronal activity, and some autistic behavior in UBE3A-overexpressing conditions. Molecular and synaptic dynamics underlying ASD pathophysiology remain unclear mechanisms and pathways, even though the glutamatergic synaptic transmission is known to be important for a part of the ASD pathophysiology. This paper contributes to uncovering the fundamental synaptic mechanisms and molecular pathways underlying ASD-related behavior, which facilitates understanding the core of the ASD mechanism. This paper would be impactful and attractive to not only molecular neuroscientists but also clinical psychiatrists working on basic research. Overall data quality looks fine to convince their claims and it is demonstrated in detail. However, there are still concerning points in the current dataset, and the author's claims. If the authors approach all of the concerns, I believe this paper would be suitable for publishing in the EMBO Reports.

Major

-In Fig. 1, the authors incubated each cultured condition with glutamate under the application of APV and TTX and found the elevated percentage of c-Fos-positive cells was abolished by UBE3A overexpression. However, the c-Fos reflects plasticity of neuronal activity, but not basal synaptic transmission. I recommend recording the mEPSC in TTX+APV following perfusing CNQX to confirm that the synaptic responses are mediated by the AMPAR here.

-Although the downregulation of Hhex, the second top candidate in the RNA-seq from the PFC, would be expected to induce the opposite phenotype to the observed localization of GluA1 mRNA, there is a possibility that indirect effects of enhanced transport of the other mRNAs can affect the GluA1 mRNA dynamics. It is recommended to perform the same analysis about Hhex as in Fig. 3 and it would strengthen the current dataset.

-Although the experimental conditions could be different between this manuscript and the previous one that Smith et al. reported previously, the frequency of the mEPSC is attenuated in the Ube3a 2xTg-transgenic mice, while there is no effect in the UBE3A-overexpressing rat primary neurons. How can this discrepancy be interpreted? One possibility is that the amount of UBE3A protein can be varied in the UBE3A-overexpressing neurons compared to the 2xTg transgenic mice. In that case, data from the UBE3A-overexpressing condition cannot be compared to that of the UBE3A 2xTg mice. To confirm a comparable amount of UBE3A expression between both conditions, it's recommended to perform the ELISA to quantify the protein level. Otherwise, the authors should discuss this discrepancy in the discussion part.

-In Fig. 5A, the authors incubated each cultured condition as performed in Fig. 1. Again, the c-Fos does not reflect basal synaptic transmission. Although the authors assess the mEPSC here, it would be better to record evoked EPSC in each condition and compare the amplitude of the AMPAR-mediated synaptic currents.

-Along with the previous point, a comparison of the decay among the conditions is recommended in the mEPSC. The lack of some subunits reflects an alteration in the kinetics of the AMPAR-mediated currents. It should be analyzed in both AMPAR-mediated mEPSC and evoked EPSC.

- The authors claimed that the SARNP-mediated pathway does not regulate memory function in Fig. S5B. Because the neonatal virus injection may not induce enough expression of SARNP in the hippocampus, it would be better to perform the virus injection in the adult age and check the object recognition.

Minor

-Throughout the entire manuscript, the terminology of gene and protein names make me confused. Mouse gene names should be like "Ube3a", while protein names are like "UBE3A" in both humans and rodents. It's better to correct all of them in the same way through the revised manuscript.

- Definition of mouse PFC is still controversial, and different among several papers. How the authors define "PFC" and "mPFC" is vague in the current manuscript. It's meaningful to clarify the definition such as "mPFC is the prelimbic cortex, infralimbic cortex, and anterior cingulate cortex".

-The way to distinguish fluorescent signals in the nucleus from that in the cytosol is unclear in Fig. 2 and Fig. 3. It would be better to describe the exact criteria for that in the method part.

We would like to thank the reviewers for their detailed feedback regarding our manuscript. We appreciate the constructive comments and suggestions. Please find the point-by-point response as below:

Referee #1:

The goal of the current research is to determine the neurobiological mechanisms underlying Ube3A-dependent autism spectrum disorders (ASD). Ube3A is known to be an E3 ubiquitin ligase and transcription coactivator. Also, the Ube3A gene is a known ASD risk gene, and increased Ube3A activity is implicated in the development of ASD in humans. The current work examines the neurobiological underpinnings of ASD in the context of Ube3A overexpression, with a particular focus on AMPA receptor (AMPA) malfunction as the cause. The study reveals that the expression of AMPARs and their synaptic distribution are significantly reduced in neurons with UBE3A overexpression and in the brains of Ube3A transgenic mice. Moreover, the team finds that the mRNA of the GluA1 subunit is restricted in the nucleus, and this mRNA retention causes a reduction in GluA1 protein synthesis. Additionally, the current research identifies that SARNP, an mRNA nuclear export protein, is downregulated in neurons overexpressing Ube3A, which is responsible for GluA1 mRNA nuclear retention. Finally, the team also confirms their findings in animal behaviors. Overall, these findings indicate that SARNP plays a crucial role in the cellular and behavioral phenotypes of Ube3A-associated ASD by regulating the nuclear mRNA trafficking and protein translation of the AMPAR GluA1 subunit. The current research topic is significant, and the integrated experimental designs are excellent. However, there are several major issues that are necessary to be addressed.

Major issues

1. Ube3a functions as a transcription coactivator. However, it is unclear how higher activity of Ube3A downregulates SARNP? The authors should **address the potential mechanisms**.

Response: We appreciate the reviewer's encouraging comments and thanks for the suggestions. We have updated the manuscript to include a discussion on potential mechanisms of Ube3a downregulating SARNP in the Discussion section:

"Under conditions of UBE3A overexpression, we detected a decrease in SARNP protein, both in vitro and in vivo. Because SARNP is not a target for UBE3A-induced ubiquitination, the decrease unlikely results from protein degradation (Fig. 2A). In fact, in addition to functioning as an E3 ubiquitin ligase, UBE3A also serves as a transcription coactivator, suggesting an involvement of transcriptional regulation. In line with this, our RNA sequencing analysis of Ube3a 2xTg brains demonstrated a reduction in SARNP mRNA level (Fig. 3A). Transcription factors can have dual functions, either activating or repressing the expression of target genes in different contexts (Boyle & Després, 2010; Latchman, 2001). It is possible that UBE3A activates the function or expression of a transcription factor which then suppresses SARNP expression. Moreover, previous research has shown that transcriptional coactivators could also suppress gene transcription under different contexts (Ma et al., 2021; Nikolenko et al., 2005; Shin et al., 2020)."

2. The authors used in vitro and in vivo transfection and infection of Ube3A and SARNP throughout the study. However, there is insufficient data to confirm whether these proteins are expressed. For instance, the authors expressed Ube3A and SARNP using AAV and lentivirus, respectively, in Figs. 4A and 5. Since

it is implausible that two separate viruses would be transduced concurrently, it is uncertain if these two proteins were indeed expressed in the same cells. The data demonstrating these expressions should be included.

Response: In Figures 4 and 5, we always infected the cultured neurons with the AAV2 virus (GFP or UBE3A) on day-in-vitro 3 (DIV3) and with lentivirus (GFP or SARNP) on DIV7, then collected the cell lysates or processed the cells for immunostaining or electrophysiological recording on DIV 11-14. AAV2 was previously shown to transduce about 85% of all cells in cultured neurons (Royo et al., 2008). Lentivirus was shown to transduce about 85-90% of cultured neurons (Ding & Kilpatrick, 2013). Therefore, both viruses should be expressed in most of the neurons. Consistent with this, in Figure 4A, we confirmed the expression of the viruses by detecting the overexpression of SARNP and UBE3A in the input cell lysates (shown on the right side), indicating robust expression of the viruses.

3. In Fig 6, it is unclear whether SARNP is indeed expressed in prefrontal and hippocampal neurons to reverse the cFos phenotype. The data confirming SARNP expression in these areas should be included.

Response: We appreciate the reviewer's suggestion. To probe the expression of the GFP-SARNP virus in prefrontal and hippocampal neurons, we collected brain lysates from the virus-injected animals. As expected, we observed endogenous SARNP expression with a molecular weight of 29kDa across all groups and the expression of the GFP-SARNP virus with a molecular weight of 54kDa in the "Ube3a 2xTg + LV-GFP-SARNP" group in both the hippocampus (HPC) and the prefrontal cortex (PFC), indicating robust expression of LV-GFP-SARNP in HPC and PFC. The figures and manuscript have been updated to incorporate these data as "Figure EV5A, B".

Figure for referee with unpublished data has been removed upon request by the authors.

4. In three chamber experiments, the authors should compare the interaction times within the experimental groups – e.g. WT vs. 2xTg vs 2xTg+SARNP – in addition to the interactions between empty and stranger and between novel and familiar.

Response: We appreciate the reviewer's feedback. In ASD studies, the relative times each genotype spends between "stranger vs. empty", or "familiar vs. novel" are compared to indicate changes in preference in social behavior (Gerasimenko et al., 2020; Huang et al., 2020; Jaramillo et al., 2017; Pan et al., 2022; Park et al., 2018; Tran et al., 2020). And thus our observations of significant changes in relative interactions between empty and stranger and between novel and familiar indicate impaired social behavior and the rescue effects of SARNP restoration.

As suggested, we also compared WT vs. 2xTg vs. 2xTg+SARNP for each category. Statistical significance was not found in these comparisons except for the interaction time with the familiar mouse between WT and 2xTg. However, as expected, the comparisons between "WT vs. 2xTg" and "2xTg vs. 2xTg+SARNP" have revealed smaller P-values.

Figure for referee with unpublished data has been removed upon request by the authors.

5. It is unclear if there is sex difference.

Response: We performed new experiments and compared sex differences. Using brain lysates from male and female WT and Ube3a 2xTg mice, we detected the expression of SARNP and GluA1 by western blotting. The results showed that both GluA1 and SARNP were reduced similarly in male and female Ube3a 2xTg mice compared to their WT counterparts, indicating a lack of sex differences in the GluA1 dysregulation and the SARNP-mediated pathway.

Figure for referee with unpublished data has been removed upon request by the authors.

6. The manuscript stated “The cellular and molecular mechanisms underlying the synaptic dysfunction in UBE3A-dependent ASD remain unclear”. However, in the previous study (Smith et al., 2011), overexpression of Ube3A in mice shows that synaptic transmission is suppressed because of reduced presynaptic release probability, synaptic glutamate concentration, and postsynaptic action potential coupling, which contributes to ASD. Therefore, how the presented mechanism integrates or acts independently from these described mechanisms should be discussed.

Response: We thank the reviewers for pointing this out. While the previous work indicated a change in presynaptic function and electrophysiological properties, our study focused on the postsynaptic dysregulation in key functional components and the molecular mechanisms. We measured the expression of postsynaptic receptors, including AMPARs, NMDARs, and GABA receptors, and provided mechanistic details on how AMPARs are reduced via SARNP-mediated mRNA redistribution, which leads to aberrant AMPAR currents. We have also confirmed the contribution of these molecular defects to the autistic phenotypes in Ube3a-dependent ASD.

We have updated the manuscript to clarify the statements and emphasize on the postsynaptic molecular mechanisms.

Minor issues

1. The previous study shows that ultrasonic vocalization is regulated by the hypothalamus (Zimmer et al., 2019). Therefore, this discussion can be included.

Response: We thank the reviewer for the suggestion. We have updated the Discussion section to include a discussion of the involvement of the hypothalamus in isolation-induced ultrasonic vocalizations in neonates:

“Isolation-induced ultrasonic vocalizations in neonates are regulated by the hypothalamus (Zimmer et al., 2019). It's possible that the LV-GFP-SARNP virus failed to induce sufficient SARNP expression in the hypothalamus. These possibilities will be clarified by future studies.”

2. Mice in the FVB background are blind. Therefore, the objective recognition test and other behavioral assessments that rely on visual cues are inappropriate for use with these animals.

Response: Thanks for noticing this phenotypic feature of this line. In fact, the FVB/NJ mouse line is one of the most social mouse lines, exhibiting significant levels of sociability and social novelty, which makes it a common genetic line for ASD mouse models (Moy et al., 2007). In the original paper that created the Ube3a 2xTg mouse line (FVB background), they tested many autism-relevant behaviors in mice at 4--16 weeks old (Smith et al., 2011) including social interaction, social communication, novel object test, open field, and grooming, and they show similar results as our findings when comparing WT+LV-GFP vs. Ube3a 2xTg+LV-GFP.

FVB/NJ mice are homozygous for the mutant phosphodiesterase 6b allele Pde6brd1 (rd1), which confers retinal degeneration. The decline of vision has been reported to begin at weaning age around P21 which can progress to blindness as early as two months of age (Farley et al., 2011). Another study suggests that

although rods completely degenerate by 9 weeks of age, cones survive up to at least 18 months of age in mice homozygous for the *rd1* mutation (Carter-Dawson et al., 1978). Our behavioral tests were performed between P30-P40 (except ultrasonic vocalization which was performed at P9), when the animals might be poor-sighted but were not blind. In addition, other sensory cues such as olfaction may be used in these tests.

3. Insufficient information about viruses (the sources, titers, and etc).

Response: We have updated the "Viral constructs preparation and virus infection" in the Method section to include description of the sources and titers of the lentiviruses:

"The pQTEV-CIP29/SARNP plasmid (Addgene, cat. #34822) was used to extract SARNP, which was then inserted into the pEGFP-C1 vector to create the pEGFP-SARNP construct. GFP-SARNP was then subcloned into pFUW lentiviral (LV) vector (Addgene, cat. #14882), to generate pFUW-GFP-SARNP construct. To package LV particles, HEK293T cells at 70% confluency in 15-cm culture plates were transfected with pFUW-GFP-SARNP, along with viral packaging and envelope proteins (pRSV/REV, pMDLg/RRE, and pVSVG) using polyethylenimine transfection reagent (Polysciences; cat. #23966). 48 h after transfection, cell medium was filtered through a 0.45 μ m filter, and PEG-it virus precipitation solution (1:5, System Biosciences; cat. # LV810A-1) was added and kept at 4°C overnight. On the next day, the solution was centrifuged (1500 rpm, 30 min, 4°C), and the pellet was resuspended with 100 μ l cold PBS. The virus was aliquoted and stored at -80°C until use. Viral particle titers were determined using qPCR with SYBR green. The titers were calculated based on the Ct values and a standard curve."

Referee #2:

The manuscript by Tian et al. explores an interesting mechanism linked to overexpression of the autism risk gene UBE3A. While the field focuses on loss of function (Angelman Syndrome), the paper summarizes in the introduction the effects of UBE3A duplication and overexpression studies. The mechanism proposed is clear and the experiments support the claims of the authors. I only have some minor points that would need to be addressed prior to publication:

1. The colocalization of GluA1 mRNA and the nuclear dye (e.g. Fig. 3H, J). This needs to be better described in the methodology and for some of the images it is not intuitive (apart from the merged image) what the significance of the dotted white circles is.

Response: We thank the reviewer for the suggestions. We have updated the "RNA fluorescent in situ hybridization" in the Method section to include a description of the measurement of nuclear GluA1 mRNA by its colocalization with the Hoechst nuclear dye. Additionally, we have revised the legends of Figure 3H, J to include a description of the Hoechst dye and the dotted circle. Basically, nuclear GluA1 mRNA was identified by its colocalization with the Hoechst dye, and the remaining signal in the soma was quantified to determine the levels of cytosolic GluA1 mRNA.

2. USV- I could not find mention of the temperature of embryos in methods during USV recording.

Response: We thank the reviewer for pointing out this problem. The pups were maintained at room temperature (~22 °C) during USV recording as suggested in previous studies (Ferhat et al., 2016; Smith et al., 2011). We have updated the “Ultrasonic vocalization (USV) recording” in the method section to include a description of the recording temperature.

3. Degradation of GluA1 by UBE3A. While the authors examine mRNA localization and protein synthesis, it would be good to discuss the potential contribution of GluA1 degradation given that the enzyme is a ubiquitin E3 ligase.

Response: In Figure 2A, we investigated whether GluA1 is a target of UBE3A when functioning as an E3 ligase. We conducted a ubiquitination assay of GluA1, and observed that UBE3A overexpression neither caused an increase in GluA1 ubiquitination levels nor led to a reduction in GluA1 protein levels, indicating that GluA1 is not a ubiquitination target of UBE3A.

Referee #3:

Tian et al reported that the expression of AMPARs and their synaptic distribution were significantly decreased in UBE3A-overexpressing neurons and in Ube3A 2xTg brains. GluA1 mRNA was restricted in the nucleus, which is regulated by SARNP, a mRNA nuclear export factor, in the overexpressing neurons. The authors also presented that restoration of SARNP levels rescued GluA1 expression, neuronal activity, and some autistic behavior in UBE3A-overexpressing conditions. Molecular and synaptic dynamics underlying ASD pathophysiology remain unclear mechanisms and pathways, even though the glutamatergic synaptic transmission is known to be important for a part of the ASD pathophysiology. This paper contributes to uncovering the fundamental synaptic mechanisms and molecular pathways underlying ASD-related behavior, which facilitates understanding the core of the ASD mechanism. This paper would be impactful and attractive to not only molecular neuroscientists but also clinical psychiatrists working on basic research. Overall data quality looks fine to convince their claims and it is demonstrated in detail. However, there are still concerning points in the current dataset, and the author's claims. If the authors approach all of the concerns, I believe this paper would be suitable for publishing in the EMBO Reports.

Major

1. In Fig. 1, the authors incubated each cultured condition with glutamate under the application of APV and TTX and found the elevated percentage of c-Fos-positive cells was abolished by UBE3A overexpression. However, the c-Fos reflects plasticity of neuronal activity, but not basal synaptic transmission. I recommend recording the mEPSC in TTX+APV following perfusing CNQX to confirm that the synaptic responses are mediated by the AMPAR here.

Response: We thank the reviewer for the constructive suggestions. We recorded AMPAR mEPSC in Figure 5 to show the impaired AMPAR currents in UBE3A-overexpression conditions. As suggested, to confirm the AMPAR component in the mEPSCs, we performed additional recordings using the same paradigm as described for Figure 5. In the presence of TTX, APV, and Bicuculline, we first recorded mEPSCs for a certain time period, and then added AMPAR antagonist CNQX (10 μ M) to the extracellular solution. As expected, addition of CNQX led to a complete abolishment of mEPSCs, confirming that the synaptic responses observed in our experiments were mediated by AMPARs.

Figure for referee with unpublished data has been removed upon request by the authors.

2. Although the downregulation of Hhex, the second top candidate in the RNA-seq from the PFC, would be expected to induce the opposite phenotype to the observed localization of GluA1 mRNA, there is a possibility that indirect effects of enhanced transport of the other mRNAs can affect the GluA1 mRNA dynamics. It is recommended to perform the same analysis about Hhex as in Fig. 3 and it would strengthen the current dataset.

Response: We agree with the reviewer's insightful comments. As suggested, we tested the expression of HHEX in WT and Ube3a 2xTg brain lysates by western blot, which indeed showed a decrease of HHEX at the protein level:

Figure for referee with unpublished data has been removed upon request by the authors.

Similar to our approach in Figure 3, we tested if replenishing HHEX under UBE3A-overexpressing conditions could restore the subcellular distribution of GluA1 mRNA. We transfected primary cultured neurons with either GFP, UBE3A, or UBE3A+HHEX. Quantifying the intensity of GluA1 mRNA revealed that cells transfected with HHEX did not restore the relative distribution of GluA1 mRNA between the cytosol and nucleus, indicating that HHEX reduction is not responsible for the observed nuclear retention of GluA1 mRNA under UBE3A-overexpression conditions:

Figure for referee with unpublished data has been removed upon request by the authors.

We've updated the figures and manuscript to include these data as "Figure EV3".

3. Although the experimental conditions could be different between this manuscript and the previous one that Smith et al. reported previously, the frequency of the mEPSC is attenuated in the Ube3a 2xTg-transgenic mice, while there is no effect in the UBE3A-overexpressing rat primary neurons. How can this discrepancy be interpreted? One possibility is that the amount of UBE3A protein can be varied in the UBE3A-overexpressing neurons compared to the 2xTg transgenic mice. In that case, data from the UBE3A-overexpressing condition cannot be compared to that of the UBE3A 2xTg mice. To confirm a comparable amount of UBE3A expression between both conditions, it's recommended to perform the ELISA to quantify the protein level. Otherwise, the authors should discuss this discrepancy in the discussion part.

Response: We thank the reviewer for pointing this out. This discrepancy could originate from the use of different preparations. Smith et al. recorded layer 2/3 neurons of the barrel cortex in brain slices from Ube3a 2xTg animals, whereas we utilized UBE3A-overexpressing cultured cortical neurons. The in vivo neural circuits and synaptic connectivity should significantly differ from the circumstances of primary neurons in vitro. Such variations might contribute to the different observations regarding the frequency of mEPSCs. We have also updated the manuscript to include a discussion of this discrepancy in the Discussion section.

4. In Fig. 5A, the authors incubated each cultured condition as performed in Fig. 1. Again, the c-Fos does not reflect basal synaptic transmission. Although the authors assess the mEPSC here, it would be better to record evoked EPSC in each condition and compare the amplitude of the AMPAR-mediated synaptic currents.

Response: We appreciate the feedback. The use of c-fos was not intended to indicate the basal synaptic transmission. Rather, we applied glutamate to activate AMPARs, which then causes neuronal activation and c-fos expression. Therefore, the level of c-fos will correlate to the abundance of membrane AMPARs. We currently don't have the setup for evoked EPSC but it will be considered in future experiments.

5. Along with the previous point, a comparison of the decay among the conditions is recommended in the mEPSC. The lack of some subunits reflects an alteration in the kinetics of the AMPAR-mediated currents. It should be analyzed in both AMPAR-mediated mEPSC and evoked EPSC.

*Response: We thank the reviewer for their constructive suggestions. As suggested, we have updated Figure 5 to include an analysis of the decay time τ (tau) in our AMPAR mEPSC recordings. We found no significant changes in the decay time τ across the four conditions, as illustrated in **Figure 5F**. These results are consistent with a previous study in which GluA1 KO neurons showed no change in the decay time τ in AMPAR mEPSCs compared to WT neurons (Toyoda et al., 2009).*

6. The authors claimed that the SARNP-mediated pathway does not regulate memory function in Fig. S5B. Because the neonatal virus injection may not induce enough expression of SARNP in the hippocampus, it would be better to perform the virus injection in the adult age and check the object recognition.

Response: We appreciate the reviewer's suggestion. In Fig. EV5D (Fig. S5B in our first submission), we observed no significant impairment in the memory function of Ube3a 2xTg animals, as the 'Ube3a 2xTg + LV-GFP' mice spent a similar amount of time exploring the novel object compared to the 'WT + LV-GFP' animals. Since the memory function is intact in Ube3a 2xTg mice, the dysregulation of the SARNP-mediated pathway observed in Ube3a 2xTg mice unlikely contributes to memory function.

Moreover, to confirm expression of the LV-GFP-SARNP virus in the hippocampus, we collected brain lysates from the virus-injected animals. We observed the endogenous SARNP at 29kd across all groups and also a strong expression of GFP-SARNP virus in "Ube3a 2xTg + LV-GFP-SARNP" group at 54kd in the hippocampus.

Figure for referee with unpublished data has been removed upon request by the authors.

Given that the memory function remains intact in both the Ube3a 2xTg mice and mice with robust expression of SARNP in the hippocampus following neonatal virus injection, SARNP doesn't seem to play a major role in memory. Nevertheless, the possibilities will be explored in our future studies.

Minor

1. Throughout the entire manuscript, the terminology of gene and protein names make me confused. Mouse gene names should be like "Ube3a", while protein names are like "UBE3A" in both humans and rodents. It's better to correct all of them in the same way through the revised manuscript.

Response: We thank the reviewer for pointing out this issue. We have updated the manuscript to correct the terminology: 'Ube3a' refers to the mouse gene. 'UBE3A' refers to the human gene. 'UBE3A' refers to mouse and human proteins.

2. Definition of mouse PFC is still controversial, and different among several papers. How the authors define "PFC" and "mPFC" is vague in the current manuscript. It's meaningful to clarify the definition such as "mPFC is the prelimbic cortex, infralimbic cortex, and anterior cingulate cortex".

Response: We thank the reviewer for the suggestion. We have updated the manuscript to include a clarification of mPFC:

"We then analyzed the medial prefrontal cortex (mPFC) that includes the prelimbic cortex, infralimbic cortex and the anterior cingulate cortex, a critical region related to social behaviors (Ko, 2017; Lee et al.,

2016; Sun et al., 2020). *We observed a decline in c-Fos expression in Ube3a 2xTg mice, which was partially restored by LV-GFP-SARNP (Fig. 6E).*”

3. The way to distinguish fluorescent signals in the nucleus from that in the cytosol is unclear in Fig. 2 and Fig. 3. It would be better to describe the exact criteria for that in the method part.

Response: We thank the reviewer for the suggestions. We have updated the “RNA fluorescent in situ hybridization” in the method section to include a description of the measurement of nuclear GluA1 mRNA by its colocalization with the nuclear dye Hoechst, as well as the legends of Figure 2 and 3 to include a description of the nuclear dye Hoechst and the dotted circle. Basically, nuclear GluA1 mRNA was assessed by its colocalization with the Hoechst nuclear dye. The remaining signal in the soma was measured to quantify the levels of cytosolic GluA1 mRNA.

References

- Boyle, P., & Després, C. (2010). Dual-function transcription factors and their entourage: Unique and unifying themes governing two pathogenesis-related genes. *Plant Signaling & Behavior*, 5(6), 629. <https://doi.org/10.4161/PSB.5.6.11570>
- Carter-Dawson, L., LaVail, M., ... R. S. & visual, & 1978, undefined. (1978). Differential effect of the rd mutation on rods and cones in the mouse retina. *Citeseer*. <https://citeseerx.ist.psu.edu/document?repid=rep1&type=pdf&doi=8d8bd092b634b2f3985d9bb01128bafd477dd500>
- Ding, B., & Kilpatrick, D. L. (2013). Lentiviral vector production, titration, and transduction of primary neurons. *Methods in Molecular Biology*, 1018, 119–131. https://doi.org/10.1007/978-1-62703-444-9_12/COVER
- Farley, S. J., McKay, B. M., Disterhoft, J. F., & Weiss, C. (2011). Reevaluating hippocampus dependent learning in FVB/N mice. *Behavioral Neuroscience*, 125(6), 871. <https://doi.org/10.1037/A0026033>
- Ferhat, A. T., Torquet, N., Le Sourd, A. M., De Chaumont, F., Olivo-Marin, J. C., Faure, P., Bourgeron, T., & Ey, E. (2016). Recording mouse ultrasonic vocalizations to evaluate social communication. *Journal of Visualized Experiments*, 2016(112), 1–12. <https://doi.org/10.3791/53871>
- Gerasimenko, M., Cherepanov, S. M., Furuhashi, K., Lopatina, O., Salmina, A. B., Shabalova, A. A., Tsuji, C., Yokoyama, S., Ishihara, K., Brenner, C., & Higashida, H. (2020). Nicotinamide riboside supplementation corrects deficits in oxytocin, sociability and anxiety of CD157 mutants in a mouse model of autism spectrum disorder. *Scientific Reports* 2020 10:1, 10(1), 1–12. <https://doi.org/10.1038/s41598-019-57236-7>
- Huang, W. C., Zucca, A., Levy, J., & Page, D. T. (2020). Social Behavior Is Modulated by Valence-Encoding mPFC-Amygdala Sub-circuitry. *Cell Reports*, 32(2). <https://doi.org/10.1016/J.CELREP.2020.107899>
- Jaramillo, T. C., Speed, H. E., Xuan, Z., Reimers, J. M., Escamilla, C. O., Weaver, T. P., Liu, S., Filonova, I., & Powell, C. M. (2017). Novel Shank3 mutant exhibits behaviors with face validity for autism and altered striatal and hippocampal function. *Autism Research*, 10(1), 42–65. <https://doi.org/10.1002/AUR.1664>
- Ko, J. (2017). Neuroanatomical substrates of rodent social behavior: The medial prefrontal cortex and its projection patterns. *Frontiers in Neural Circuits*, 11, 41. <https://doi.org/10.3389/FNCIR.2017.00041/BIBTEX>

- Latchman, D. S. (2001). Transcription factors: Bound to activate or repress. *Trends in Biochemical Sciences*, 26(4), 211–213. [https://doi.org/10.1016/S0968-0004\(01\)01812-6](https://doi.org/10.1016/S0968-0004(01)01812-6)
- Lee, E., Rhim, I., Lee, J. W., Ghim, J. W., Lee, S., Kim, E., & Jung, M. W. (2016). Enhanced Neuronal Activity in the Medial Prefrontal Cortex during Social Approach Behavior. *Journal of Neuroscience*, 36(26), 6926–6936. <https://doi.org/10.1523/JNEUROSCI.0307-16.2016>
- Ma, L., Chen, S., Wang, Z., Guo, S., Zhao, J., Yi, D., Li, Q., Liu, Z., Guo, F., Li, X., Jia, P., Ding, J., Liang, C., & Cen, S. (2021). The CREB Regulated Transcription Coactivator 2 Suppresses HIV-1 Transcription by Preventing RNA Pol II from Binding to HIV-1 LTR. *Virologica Sinica*, 36(4), 796. <https://doi.org/10.1007/S12250-021-00363-1>
- Moy, S. S., Nadler, J. J., Young, N. B., Perez, A., Holloway, L. P., Barbaro, R. P., Barbaro, J. R., Wilson, L. M., Threadgill, D. W., Lauder, J. M., Magnuson, T. R., & Crawley, J. N. (2007). Mouse Behavioral Tasks Relevant to Autism: Phenotypes of Ten Inbred Strains. *Behavioural Brain Research*, 176(1), 4. <https://doi.org/10.1016/J.BBR.2006.07.030>
- Nawaz, Z., Lonard, D. M., Dennis, A. P., Smith, C. L., & O'Malley, B. W. (1999). Proteasome-dependent degradation of the human estrogen receptor. *Proceedings of the National Academy of Sciences of the United States of America*, 96(5), 1858–1862. <https://doi.org/10.1073/PNAS.96.5.1858>
- Nawaz, Z., Lonard, D. M., Smith, C. L., Lev-Lehman, E., Tsai, S. Y., Tsai, M.-J., & O'Malley, B. W. (1999). The Angelman syndrome-associated protein, E6-AP, is a coactivator for the nuclear hormone receptor superfamily. *Molecular and Cellular Biology*, 19(2), 1182–1189. <https://doi.org/10.1128/MCB.19.2.1182>
- Nikolenko, J. V., Shidlovskii, Y. V., Lebedeva, L. A., Krasnov, A. N., Georgieva, S. G., & Nabirochkina, E. N. (2005). Transcriptional coactivator SYP can suppress transcription in heterochromatin. *Russian Journal of Genetics*, 41(8), 840–843. <https://doi.org/10.1007/S11177-005-0169-7/METRICS>
- Pan, L., Zheng, L., Wu, X., Zhu, Z., Wang, S., Lu, Y., He, Y., Yang, Q., Ma, X., Wang, X., Yang, H., Zhan, L., Luo, Y., Li, X., Zhou, Y., Wang, X., Luo, J., Wang, L., Duan, S., & Wang, H. (2022). A short period of early life oxytocin treatment rescues social behavior dysfunction via suppression of hippocampal hyperactivity in male mice. *Molecular Psychiatry* 2022 27:10, 27(10), 4157–4171. <https://doi.org/10.1038/s41380-022-01692-7>
- Park, M. J., Seo, B. A., Lee, B., Shin, H. S., & Kang, M. G. (2018). Stress-induced changes in social dominance are scaled by AMPA-type glutamate receptor phosphorylation in the medial prefrontal cortex. *Scientific Reports* 2018 8:1, 8(1), 1–14. <https://doi.org/10.1038/s41598-018-33410-1>
- Ramamoorthy, S., Dhananjayan, S. C., Demayo, F. J., & Nawaz, Z. (2010). Isoform-Specific Degradation of PR-B by E6-AP Is Critical for Normal Mammary Gland Development. *Molecular Endocrinology*, 24(11), 2099–2113. <https://doi.org/10.1210/ME.2010-0116>
- Ramamoorthy, S., & Nawaz, Z. (2008). E6-associated protein (E6-AP) is a dual function coactivator of steroid hormone receptors. *Nuclear Receptor Signaling*, 6. <https://doi.org/10.1621/NRS.06006>
- Royo, N. C., Vandenberghe, L. H., Ma, J. Y., Hauspurg, A., Yu, L. Y., Maronski, M., Johnston, J., Dichter, M. A., Wilson, J. M., & Watson, D. J. (2008). Specific AAV Serotypes Stably Transduce Primary Hippocampal and Cortical Cultures with High Efficiency and Low Toxicity. *Brain Research*, 1190(1), 15. <https://doi.org/10.1016/J.BRAINRES.2007.11.015>
- Shin, J. H., Lee, G., Jeong, M. G., Kim, H. K., Won, H. Y., Choi, Y., Lee, J. H., Nam, M., Choi, C. S., Hwang, G. S., & Hwang, E. S. (2020). Transcriptional coactivator with PDZ-binding motif suppresses the expression of steroidogenic enzymes by nuclear receptor 4 A1 in Leydig cells. *The FASEB Journal*, 34(4), 5332–5347. <https://doi.org/10.1096/FJ.201900695RRR>

- Smith, S. E. P., Zhou, Y. D., Zhang, G., Jin, Z., Stoppel, D. C., & Anderson, M. P. (2011). Increased Gene Dosage of Ube3a Results in Autism Traits and Decreased Glutamate Synaptic Transmission in Mice. *Science Translational Medicine*, 3(103), 103ra97. <https://doi.org/10.1126/SCITRANSLMED.3002627>
- Sun, Q., Li, X., Li, A., Zhang, J., Ding, Z., Gong, H., & Luo, Q. (2020). Ventral Hippocampal-Prefrontal Interaction Affects Social Behavior via Parvalbumin Positive Neurons in the Medial Prefrontal Cortex. *Science*, 23(3), 100894. <https://doi.org/10.1016/J.ISCI.2020.100894>
- Toyoda, H., Zhao, M. G., Ulzhöfer, B., Wu, L. J., Xu, H., Seeburg, P. H., Sprengel, R., Kuner, R., & Zhou, M. (2009). Roles of the AMPA receptor subunit GluA1 but not GluA2 in synaptic potentiation and activation of ERK in the anterior cingulate cortex. *Molecular Pain*, 5. https://doi.org/10.1186/1744-8069-5-46/ASSET/IMAGES/LARGE/10.1186_1744-8069-5-46-FIG8.JPEG
- Tran, D. N., Jung, E. M., Yoo, Y. M., Lee, J. H., & Jeung, E. B. (2020). Perinatal Exposure to Triclosan Results in Abnormal Brain Development and Behavior in Mice. *International Journal of Molecular Sciences* 2020, Vol. 21, Page 4009, 21(11), 4009. <https://doi.org/10.3390/IJMS21114009>

Dear Dr. Man,

Thank you for the submission of your revised manuscript. We have now received the enclosed comments and cross-comments from the referees who were asked to assess it.

While referee 3 is satisfied with the revised study, referee 1 is not and is asking for more data. I think all remaining comments by referee 1 should be addressed. If you already have the data regarding her/his point 1, please explain this in your point-by-point response. Please note that the referee is asking for a confirmation "that the majority of cells are positive for both SARNP and UBE3A". Regarding point 3, you can either remove the data, or discuss the data appropriately.

A few more editorial requests will also need to be addressed:

- Please add up to 5 keywords to your ms file.
- Please correct the conflict of interest subheading to "Disclosure and Competing Interests Statement"
- Please remove the author credits from the ms file. All credits need to be entered in our online submission system.
- A callout is missing for Appendix Figure S1, please add.
- Please upload the Appendix as a pdf file including Figure S1. The Appendix figure legend should be removed from the ms file and included in the Appendix file.
- A video is mentioned in the ms file but no video is uploaded.
- The text in the synopsis image is a little too small at the final image size of 550 pixels x 300 pixels. It would be better if you could send us a new image file at the correct size and with larger text.
- In addition to the bullet points, please also send us a short summary of your key results and their significance (1-2 sentences).
- The manuscript sections should be in the following order: Title page - Abstract & Keywords - Introduction - Results - Discussion - Materials & Methods - Data Availability - Acknowledgments - Disclosure Statement & Competing Interests - References - Figure Legends - Tables with legends - Expanded View Figure Legends. Please correct.
- Please note that a separate 'Data Information' section is required in the legends of all the figures. Please note that information related to "n" is missing in the legend of figure 1b.

I would like to suggest some changes to the title and abstract that needs to be written in present tense. Please let me know whether you agree with the following:

UBE3A-caused mRNA nuclear retention reduces AMPA receptor expression and neuronal activity causing autistic-like behaviors

UBE3A is a common genetic factor in ASD etiology, and transgenic mice overexpressing UBE3A exhibit typical autistic-like behaviors. Because AMPA receptors (AMPA receptors) mediate most of the excitatory synaptic transmission in the brain, and synaptic dysregulation is considered one of the primary cellular mechanisms in ASD pathology, we investigate here the involvement of AMPARs in UBE3A-dependent ASD. We show that expression of the AMPAR GluA1 subunit is decreased in UBE3A-overexpressing mice, and that AMPAR-mediated neuronal activity is reduced. GluA1 mRNA is trapped in the nucleus of UBE3A-overexpressing neurons, suppressing GluA1 protein synthesis. Also, SARNP, an mRNA nuclear export protein, is downregulated in UBE3A-overexpressing neurons, causing GluA1 mRNA nuclear retention. Restoring SARNP levels not only rescues GluA1 mRNA localization and protein expression, but also normalizes neuronal activity and autistic behaviors in mice overexpressing UBE3A. These findings indicate that SARNP plays a crucial role in the cellular and behavioral phenotypes of UBE3A-induced ASD by regulating nuclear mRNA trafficking and protein translation of a key AMPAR subunit.

I look forward to seeing a final version of your manuscript when it is ready.

Esther Schnapp, PhD
Senior Editor

Referee #1:

Although the revised manuscript has been significantly improved, some issues still remain.

1. The study (Kim et al., Eur J Neurosci. 2014) has shown that co-injection of two AAVs (the same promotor) of different serotypes yields a largely non-overlapping pattern of viral expression in the brain. In the current study, the authors infected two different viruses (seemingly two different promotors) at different time points. Therefore, it is essential to show that UBE3A and SARNP are indeed co-infected into the same cells. Although the authors show the expression UBE3A and SARNP using immunoblots in Figs. 4 and 5, whether the expression is from the same cells is still unknown. It is still possible that they can be mutually exclusive in their expression in cultures. To confirm that the majority of cells are positive for both SARNP and UBE3A, immunostaining should be done.
2. In Fig. 6, the authors only provided immunoblots to show SARNP expression. However, it is unclear whether SARNP is expressed in neurons that are also positive c-Fos and whether SARNP expression signals are from non-neuronal cells. Therefore, immunohistochemistry showing that the majority of neurons are positive for both SARNP and c-Fos is required.
3. The novel object recognition test is known to mainly involve visual exploration of two objects. A large volume of studies has suggested that FVB mice' learning abilities being deeply impaired in spatial tasks relying on visual stimuli such as the Morris water maze and radial maze (Brown et al., Learning Memory, 2007, Owen et al., Neuroscience, 1997, Pugh et al., Behav. Brain Res. 2004, Royle et al., Brain Res. 1999, Volkmar et al., Physiol. Behv. 2001, Mineur et al., Brain Res. 2002, Girard et al., Behav. Brain Res. 2016). Additionally, the study (Farley et al., 2011) referred by the authors in the response had used hippocampal dependent tasks when the need for visual cues is minimized. Therefore, the behavioral tests that are heavily dependent on visual cues are not suitable for FVB mice although they are not completely blind as the authors argued. Without proper controls and comprehensive visual analysis in these mice, inclusion of the novel object recognition test data may mislead the research field.
4. The information of viruses (e.g. serotype, titers, and promotors) is still insufficient.

Cross-comments from referee 3:

On comment No.1, I think the authors have already shown that the immunoblots data in Fig. 4, which is good enough to answer the question.

On the second, I agree with reviewer 1's opinion. It might be better to show colocalization between GFP (SARNP) and c-fos.

In the last one, I do not think what reviewer 1 mentioned is necessary in the current manuscript.

Further comments from referee 1:

Regarding point 3, since it is supplementary, and learning and memory is not directly linked to autism, I suggest removing the data.

Referee #1:

We thank the reviewer for the constructive comments and suggestions.

1. The study (Kim et al., Eur J Neurosci. 2014) has shown that co-injection of two AAVs (the same promotor) of different serotypes yields a largely non-overlapping pattern of viral expression in the brain. In the current study, the authors infected two different viruses (seemingly two different promotors) at different time points. Therefore, it is essential to show that UBE3A and SARNP are indeed co-infected into the same cells. Although the authors show the expression UBE3A and SARNP using immunoblots in Figs. 4 and 5, whether the expression is from the same cells is still unknown. It is still possible that they can be mutually exclusive in their expression in cultures. To confirm that the majority of cells are positive for both SARNP and UBE3A, immunostaining should be done.

Response: We infected primary cultured neurons with LV-GFP-SARNP and AAV-Myc-UBE3A using the methods described in Figure 4 and Figure 5. The results showed a high level of co-localization of GFP-SARNP and Myc-UBE3A expression in the neurons.

Figure for referee with unpublished data has been removed upon request by the authors.

2. In Fig. 6, the authors only provided immunoblots to show SARNP expression. However, it is unclear whether SARNP is expressed in neurons that are also positive c-Fos and whether SARNP expression signals are from non-neuronal cells. Therefore, immunohistochemistry showing that the majority of neurons are positive for both SARNP and c-Fos is required.

Response: We thank the reviewer for the comment. We re-imaged the Ube3a 2xTg brain slices infected with LV-GFP-SARNP by immunohistochemistry to include the GFP channel. We further examined our imaging data and observed that most neurons positive for GFP-SARNP were indeed also positive for c-Fos in both the hippocampus and the prefrontal cortex. We have also added these data to Figure EV5 (Scale bars, 500 μ m).

Figure for referee with unpublished data has been removed upon request by the authors.

3. The novel object recognition test is known to mainly involve visual exploration of two objects. A large volume of studies has suggested that FVB mice' learning abilities being deeply impaired in spatial tasks relying on visual stimuli such as the Morris water maze and radial maze (Brown et al., Learning Memory, 2007, Owen et al., Neuroscience, 1997, Pugh et al., Behav. Brain Res. 2004, Royle et al., Brain Res. 1999, Volkar et al., Physiol. Behv. 2001, Mineur et al., Brain Res. 2002, Girard et al., Behav. Brain Res. 2016). Additionally, the study (Farley et al., 2011) referred by the authors in the response had used hippocampal

dependent tasks when the need for visual cues is minimized. Therefore, the behavioral tests that are heavily dependent on visual cues are not suitable for FVB mice although they are not completely blind as the authors argued. Without proper controls and comprehensive visual analysis in these mice, inclusion of the novel object recognition test data may mislead the research field.

Response: Thanks for the insight. Following the reviewer's suggestion, we have removed the figure and text related to novel object test.

4. The information of viruses (e.g. serotype, titers, and promoters) is still insufficient.

Response: We have added a section about the AAV-UBE3A virus and more information about the pFUW-GFP-SARNP lentivirus in the 'Viral Constructs Preparation and Virus Infection' section of the Methods.

Dr. Hengye Man
Boston University
Biology
5 Cummington Mall
Boston, MA 02215
United States

Dear Dr. Man,

I am very pleased to accept your manuscript for publication in the next available issue of EMBO reports. Thank you for your contribution to our journal.

I would like to suggest to slightly modify the title and abstract to:

mRNA nuclear retention reduces AMPAR expression and promotes autistic behavior in UBE3A-overexpressing mice

UBE3A is a common genetic factor in ASD etiology, and transgenic mice overexpressing UBE3A exhibit typical autistic-like behaviors. Because AMPA receptors (AMPA receptors) mediate most of the excitatory synaptic transmission in the brain, and synaptic dysregulation is considered one of the primary cellular mechanisms in ASD pathology, we investigate here the involvement of AMPARs in UBE3A-dependent ASD. We show that expression of the AMPAR GluA1 subunit is decreased in UBE3A-overexpressing mice, and that AMPAR-mediated neuronal activity is reduced. GluA1 mRNA is trapped in the nucleus of UBE3A-overexpressing neurons, suppressing GluA1 protein synthesis. Also, SARNP, an mRNA nuclear export protein, is downregulated in UBE3A-overexpressing neurons, causing GluA1 mRNA nuclear retention. Restoring SARNP levels not only rescues GluA1 mRNA localization and protein expression, but also normalizes neuronal activity and autistic behaviors in mice overexpressing UBE3A. These findings indicate that SARNP plays a crucial role in the cellular and behavioral phenotypes of UBE3A-induced ASD by regulating nuclear mRNA trafficking and protein translation of a key AMPAR subunit.

Please let me know in case you do not agree with these changes, and we can discuss this further. Otherwise we will proceed with this modified text.

Yours sincerely,
